# The potential of [230]Th for detection of ocean acidification impacts on pelagic carbonate production

Christoph Heinze[1,2], Tatiana Ilyina[3], and Marion Gehlen[4]

[1]Geophysical Institute, University of Bergen and Bjerknes Centre for Climate Research, Bergen, 5020, Norway
[2]Uni Research Climate, Bergen, 5007, Norway
[3]Max Planck Institute for Meteorology, Hamburg, 20146, Germany
[4]Laboratoire des Sciences du Climat et de l'Environnement UMR CEA-CNRS-UVSQ, Gif-sur-Yvette, 91191, France

*Correspondence to*: Christoph Heinze (christoph.heinze@uib.no)

**Abstract.** Concentrations of dissolved [230]Th in the ocean water column increase with depth due to scavenging and downward particle flux. Due to the [230]Th scavenging process, any change in the calcium carbonate ($CaCO_3$) fraction of the marine particle flux due to changes in biological $CaCO_3$ hard shell production as a consequence of progressing ocean acidification would be reflected in the dissolved [230]Th activity. Our prognostic simulations with a biogeochemical ocean general circulation model using different scenarios for the reduction of $CaCO_3$ production under ocean acidification and different greenhouse gas emission scenarios (RCPs 8.5 to 2.6) reveal the potential for deep [230]Th measurements to detect reduced $CaCO_3$ production at the sea surface. The time of emergence of an acidification induced signal on dissolved [230]Th is of the same order of magnitude as for alkalinity measurements. Interannual and decadal variability in other factors than a reduction in $CaCO_3$ hard shell production may mask the ocean acidification induced signal in dissolved [230]Th and make detection of the pure $CaCO_3$ induced signal more difficult so that only really strong changes in marine $CaCO_3$ export would be unambiguously identifiable soon. Nevertheless, the impact of changes in $CaCO_3$ export production on marine [230]Th are stronger than those for changes in POC (particulate organic carbon) or clay fluxes.

## 1 Introduction

Progressing ocean acidification is a fact. It can be directly seen from measurements at Eulerian time series stations (e.g., Bates, 2007; Dore et al., 2009; Santana-Casiano et al., 2007) and also at larger depth from high quality deep hydrography data (Olafsson et al., 2009). Depending on the emission scenario for $CO_2$, the decrease in ocean pH and the decrease in carbonate saturation will continue and become more pressing during this century (e.g., Orr et al., 2005; Steinacher et al., 2009; Bopp et al., 2013). Temporal and regional patterns of changes in pH and carbonate saturation are relatively straightforwardly to project by Earth system models including marine inorganic carbon chemistry formulations (e.g. Bopp et al., 2013). They also can be monitored through long-term high quality measurements of the inorganic carbon system. However, monitoring impacts of ocean acidification on biological processes remains challenging. While for some organisms, especially corals (Kleypas et al., 1999), detrimental effects due to decreasing pH and carbonate saturation have been determined, the various physiological impacts of ocean acidification on specific organisms and ecosystem functioning are still under investigation (e.g., Iglesias-Rodriguez et al., 2008; Kroeker et al., 2013; Meyer and Riebesell, 2015; Riebesell et al., 2007). At present, a series of possible pH-dependencies of governing marine carbon cycle parameters (such as elemental stoichiometric ratios and nutrient uptake kinetics under biological particle production) are discussed. So far, the potential decrease in

calcification due to the lowering of carbonate saturation under high $pCO_2$ is among the key changes which may be expected to occur, especially when it comes to organisms building aragonite shells (the meta-stable form of $CaCO_3$, calcite has a lower solubility than aragonite) (Raven et al., 2005). Two important questions emerge: 1. If changes in biological calcification would indeed occur in the ocean during the coming years – how can they be detected and monitored by observational frameworks (through which methodology and through which variable)? 2. In which oceanic region could one observe early signals of these changes unambiguously at the earliest possible stage?

Development of such early warning systems is vital in order to check the validity of parameterisations of pH-dependent processes in ocean models and to take appropriate mitigation/adaptation measures to diminish the consequences of potential considerable ecosystem changes due to ocean acidification. Such changes could affect the marine food chain. In a global modelling study, Ilyina et al. (2009) quantified the detection thresholds for changes in alkalinity due to changes in a series of possible formulations for the reduction of calcification (with the term "calcification" we mean here the production of $CaCO_3$ hard parts by marine biota) with pH decrease. The logic behind the approach of Ilyina et al. (2009) is as follows: If ocean acidification leads to a decrease in calcification, more $CO_3^{2-}$ ions would be retained in the surface water and not become incorporated into $CaCO_3$ shell material. Changes in $CO_3^{2-}$ ion concentration would induce a change in ocean total alkalinity, which could eventually be measured against an earlier baseline. According to that study, the tropical Pacific (with high $CaCO_3$ production rates) would be the region for detecting such alkalinity changes first, as the anticipated change in $CaCO_3$ production would be largest there in absolute terms. For intermediate dependencies of $CaCO_3$ production on pH/carbonate saturation, a reduction in biological calcification could unequivocally only be diagnosed from ca. year 2035 on. In the Arctic Ocean, where pH changes are expected to be most pronounced, large scale changes in calcification would be detectable even later on the basis of alkalinity measurements due to the overall lower biological production rates. Likewise, detection may be additionally complicated due to signals from natural seasonal and interannual variations in surface total alkalinity (Carter et al., 2016). There is thus a need for the development of novel detection methods. Heinze et al. (2006) investigated the impact of varying rain ratios $CaCO_3:C_{org}$ on the distribution of the radionuclides thorium ($^{230}$Th), protactinium ($^{231}$Pa), as well as beryllium ($^{10}$Be). As the rain ratio describes the average amount of carbon atoms incorporated into $CaCO_3$ shell material relative to the amount of carbon atoms incorporated into organic matter by plankton, a reduction in calcification would lead to a rain ratio decrease. Indeed, especially for $^{230}$Th, due to its affinity for scavenging to $CaCO_3$ particles, a considerable increase in the concentration of dissolved $^{230}$Th with depth and time was obtained in a sensitivity experiment with strong rain ratio reduction. In this paper here, we explore the option of using radionuclides for diagnosing changes in calcification and respectively reduced downward marine particle flux in more detail.

## 2 The concept - radionuclides and particles

We focus here on $^{230}$Th, a long-lived radionuclide (half-life $7.5 \cdot 10^4$ yr) and a highly particle reactive metal. $^{230}$Th is produced in the uranium decay series. As uranium has a very long residence time in the ocean and quasi-homogenous concentrations in seawater, the present natural marine $^{230}$Th source is constant everywhere in the

ocean water column and thus ideally suited for using it as a tracer in modelling studies. [230]Th is strongly particle reactive and is removed from the water column quickly through scavenging by the vertical particle flux in the ocean (for a summary, please, see Henderson et al., 1999). The majority of [230]Th is removed from the water column and will not re-enter the bottom waters, e.g., through sediment pore water diffusion (though resuspension may potentially cause some re-release). In spite of its strong particle reactivity, concentrations of dissolved [230]Th show horizontal as well as vertical gradients in the water column which are induced by the oceanic current field and differential particle fluxes as well as particle concentrations (e.g., Henderson et al., 1999; Yu et al., 1996). The distribution of particle bound [230]Th follows the particle concentrations and particle fluxes. Both, the dissolved [230]Th and the particle bound [230]Th show nutrient type vertical profiles with low values at the surface and increasing concentrations with depth. Dissolved [230]Th increases with depth as particles continuously carry [230]Th downwards and new equilibria between the dissolved phase and the particle-attached phase establish as illustrated in Figure1. The equilibrium between the concentration of the dissolved phase [[230]Th$_{diss}$] and the concentration of the particle-attached phase [[230]Th$_{part}$] can be described in an analog way to a chemical reaction equation:

$$^{230}Th_{diss} \quad \leftrightarrow \quad ^{230}Th_{part}$$

The respective analog for the mass action law constant describing to which extent the reaction from left to right is carried out is then given through the $k_d$ value governing the equilibrium between the dissolved and particle bound phases for the radionuclide:

$$k_d \; = \; \frac{\left[^{230}Th_{part}\quad\right]}{\left[^{230}Th_{diss}\quad\right]}$$

Often, $k_d$ values are formulated to account for a shift of the equilibrium towards the particle-attached phase at low particle mass concentrations assuming that low particulate concentrations occur in parallel to low particle sizes with correspondingly high reactive surface areas when compared to large particles (e.g. Honeyman et al., 1988). Respective formulations have been used successfully in [230]Th modelling studies (Heinze et al., 2006; Henderson et al., 1999).

How would then the distribution of dissolved [230]Th in the ocean reflect changes in marine calcification at the sea surface? There are indications, that [230]Th is attached first of all to $CaCO_3$ and clay particles in the water column as preferential carrier phases (Chase et al., 2002, 2003; Luo and Ku, 1999, Hayes et al., 2015b). Indeed, Heinze et al. (2006) reproduced the large-scale [230]Th distribution in the world ocean with a coarse resolution biogeochemical ocean general circulation model based on a formulation allowing [230]Th scavenging only by $CaCO_3$ and clay particles. Other studies have indicated that preferential carrier phases may vary regionally in the ocean (Scholten et al., 2005). However, Heinze et al. (2006) could demonstrate that rain ratio changes ($CaCO_3$:Corg) in marine biological particle export production could also be recorded, if [230]Th became in addition attached to particulate organic carbon (POC). A weakening of $CaCO_3$ particle production would result in a less efficient [230]Th scavenging as less particles (in terms of mass) would be available to carry [230]Th to larger depths and finally the sediment. Changes in the strength of $CaCO_3$ production and the respective downward particle flux are reflected increasingly better in the [230]Th distribution with increasing depth (see Heinze et al., 2006) due to two reasons. First, $CaCO_3$ particles get less well degraded as POC (which is remineralised mostly in the upper 1000 m of the water column) and thus reach larger depth; this is also corroborated from sediment trap measurements (e.g., (Honjo, 1996)). Second, due to the scavenging of [230]Th by particles, the vertical downward particle flux, and the equilibration

between dissolved and particle bound phases, $^{230}$Th is continuously transferred from shallower layers to larger depths. Thus temporal changes in $^{230}$Th scavenging in the upper ocean become enlarged in the deep $^{230}$Th distribution as in a kind of "magnifying glass" (see Figure 1). We investigate here, whether this feature can be exploited for an early detection method of large-scale reductions in calcification and correspondingly reduced rates in CaCO$_3$ particle export and CaCO$_3$ particle fluxes through the water column.

## 3 Model description

In this study, we use the Hamburg ocean carbon cycle circulation model HAMOCC (Maier-Reimer, 1993) in its annually averaged version (time step 1 yr, Heinze and Maier-Reimer, 1999; Heinze et al., 2009; Heinze et al., 2016) with a horizontal resolution of 3.5°x3.5°. This coarse resolution model is computationally very efficient and useful when multiple integrations are needed such as for the testing and adjusting of scavenging codes. An advantage of this fast model is that it includes a fully equilibrated early diagenesis module (10 layers) under each grid point and thus can account for alkalinity changes induced by dissolution of CaCO$_3$ from the seafloor. The model version employed here corresponds to the version as used in Heinze et al. (2009) and Heinze et al. (2016) with a slightly updated scavenging module of Heinze et al. (2006) with resepct to the formulation of the equilibrium coefficient governing the distribution of $^{230}$Th between the dissolved and particle attached phase (see below). For details, please, see these publications. The model uses a fixed ocean velocity field (and thus provides no dynamical computation of the ocean currents; velocities, temperature, salinity and ice cover are rather read from an input file). The model includes an atmospheric compartment ("slab atmosphere") which allows for prognostic computation of the atmospheric CO$_2$ concentration as well as meridional atmospheric CO$_2$ transport. We describe here only briefly features of specific relevance for this study. The water column is structured into 11 layers (centred at 25, 75, 150, 250, 450, 700, 1000, 2000, 3000, 4000, and 5000 m). The bioturbated top sediment zone of the early diagenesis module is divided into 10 layers which are separated by interfaces at 0, 0.3, 0.6, 1.1, 1.6, 2.1, 3.1, 4.1, 5.1, 7.55, and 10 cm "downcore." We make the simplifying assumption that no pore water reactions take place below 10 cm depth in the sediment (see, e.g., Smith and Rabouille, 2002; Boudreau, 1997). The biogeochemical model includes the processes of air-sea gas exchange, biogenic particle export production out of the ocean surface layer, particle flux through the water column and particle degradation by dissolution as well as remineralisation, transport of dissolved substances with the ocean currents, deposition of particulate constituents on the ocean floor, pore water chemistry and diffusion, advection of solid sediment weight fractions (organic carbon, organic phosphorus, CaCO$_3$, opal, and clay), bioturbation, and sediment accumulation (export out of the sediment mixed layer). The model predicts the following tracer concentrations in the atmosphere, the ocean water column and in the sediments. Atmospheric tracers include the concentrations of CO$_2$ (carbon dioxide) and O$_2$. In the water column, concentrations of DIC (dissolved inorganic carbon), POC (particulate organic carbon), POP (particulate organic phosphorus), DOC (dissolved organic carbon), CaCO$_3$ (calcium carbonate or particulate inorganic carbon), dissolved oxygen O$_2$, dissolved PO$_4^{3-}$ as biolimiting nutrient, silicic acid Si(OH)$_4$ and opal (biogenic particulate silica BSi) are calculated. In the sediment pore waters, the same dissolved substances as in the water column, as well as solid sediment constituents such as clay, CaCO$_3$, opal, and organic carbon are simulated. The inorganic carbon chemistry is computed following Dickson et al. (2007). In the advection scheme and for the other chemical reactions, DIC and TAlk are used as "master tracers" form which derived quantities

such as the $CO_3^{2-}$ concentration and the pH value are computed through a Newton-Raphson algorithm. In the annually averaged model as employed in this study, only export production of biogenic particles is modelled (and no explicit phytoplankton and zooplankton concentrations). Particle production takes place in the model surface layer representing the euphotic zone. Phosphate serves as biolimiting nutrient. POC and opal export productions are simulated following Michaelis Menten kinetics for nutrient uptake (e.g., Sarmiento and Gruber, 2006) (where the phytoplankton concentration is replaced by the phosphate concentration as ecosystem processes as such are not explicitly modelled):

$$P_{POC} = \frac{V_{\max}^{POC} \cdot [PO_4^{3-}]^2}{K_s^{POC} + [PO_4^{3-}]} \; ;$$

and

$$P_{opal} = \frac{V_{\max}^{opal} \cdot [Si(OH)_4]^2}{K_s^{PopalC} + [Si(OH)_4]} \; ;$$

where $P_{POC}$ and $P_{opal}$ are the POC and opal export production rates (mol l$^{-1}$ yr$^{-1}$), Red(C:P) is the Redfield ratio C:P, $V_{\max}^{POC}$ and $V_{\max}^{opal}$ are the maximum uptake rate of phosphate and silicic acid from the water column (yr-1), and $K_s^{POC}$ as well as $K_s^{opal}$ are the respective half saturation constants. $V_{\max}^{POC}$, $V_{\max}^{opal}$, $K_s^{POC}$, and $K_s^{opal}$ are simulated as a function of sea surface temperature as described by Heinze et al. (2003). POP production follows POC production with a constant stoichiometry here. The export production of $CaCO_3$ is coupled to the local production ratio $P_{opal}/P_{POC}$. It starts to increase gradually (parameter R see below) if $P_{opal}/P_{POC}$ sinks below a threshold value $S_{opal}$, i.e., when not enough silicic acid is available in the ocean surface layer to fuel full diatom growth:

$$P_{CaCO3} = P_{POC} \cdot R \cdot a \cdot \left(1 - \frac{\frac{P_{opal}}{P_{POC}}}{S_{opal}}\right) for \; \frac{P_{opal}}{P_{POC}} < S_{opal}; \quad (1)$$

$$P_{CaCO3} = 0 \; for \; \frac{P_{opal}}{P_{POC}} \geq S_{opal} \; .$$

Parameter R is the maximum possible rain ratio C(CaCO$_3$):C(POC), a is the CaCO$_3$ saturation dependent factor to account for an ocean acidification impact (see Figure 2, following Ilyina et al., 2009), and $S_{opal}$ is the threshold value of $P_{opal}/P_{POC}$ for gradual onset of CaCO$_3$ production. Particle fluxes and particle degradation are simulated through balance equations for sinking particulate matter as in Heinze et al. (2009) and Heinze et al. (2016) taking the saturation state for CaCO$_3$ and biogenic silica into account.

Scavenging of $^{230}$Th is simulated according the reversible first-order scavenging reaction (Gehlen et al., 2003; Heinze et al., 2006):

$$\frac{dc_{part}}{dt} = K \cdot [c_{part}^{EQ} - c_{part}]; \quad c_{part}^{EQ} = k_d \cdot c_{diss} \cdot M;$$

$c_{part}$ is the concentration of particle bound $^{230}$Th, $c_{diss}$ the concentration of dissolved $^{230}$Th, $c_{part}^{EQ}$ the equilibrium concentration of particle bound $^{230}$Th. The first-order rate constant K (in [yr$^{-1}$]) is set here to $10^4$ yr$^{-1}$ thus assuming a quasi-instantaneous equilibration. Concentration of suspended particulate material is represented by M. For the partitioning coefficient $k_d$ between the dissolved and particle attached phases of $^{230}$Th we follow the formulation

of (Honeyman et al., 1988), which accounts implicitly for the changing reactive surfaces of particles with particle size:

$$log_{10} k_d^{first\ guess} = A + B \cdot log_{10} M$$

where M is the particle concentration (here taken in mg particles per litre) and A as well as B are tuneable parameters. In addition to $^{230}$Th we carry also $^{231}$Pa as well as $^{10}$Be in our model (see Heinze et al., 2006), but focus here on $^{230}$Th only. For each of these radionuclides, the preferential carrier phase can be selected in a dedicated switchboard. For $^{230}$Th scavenging, we used here $CaCO_3$, POC, and clay as carrier phases. With introducing a particle specific scavenging following Hayes et al. (2015b), the final formulation for the scavenging equilibrium coefficient then becomes:

$$k_d = \left( C_{CaCO3} \cdot F_{CaCO3} + C_{POC} \cdot F_{POC} + C_{clay} \cdot F_{clay} \right) \cdot k_d^{first\ guess}$$

Where C and F are the weighting coefficients and fractions of total local particle mass. The *A*, *B*, and *C* values are included in Table 1. Atlantic and Pacific cross sections with the $k_d$ values for the control simulation without $CO_2$ emissions is given in Figure S1 (in the Supplementary Material).

The clay flux is computed according to the modern dust deposition from Mahowald et al. (1999) assuming that respective clay particles are chemically quasi-inert in seawater. Particle bound $^{230}$Th phases never get to zero, even if all biogenic particles may get degraded, because at each grid point there is – at least a tiny – dust flux consisting of inert clay. Therefore, particle concentrations as such never go to zero (a minimum concentration could be prescribed, but this was not necessary in our case). Because the Atlantic dust deposition may be overestimated (Gehlen et al., 2003), we assumed for the $^{230}$Th scavenging only 25% of the respective clay flux strength in order to avoid too strong scavenging in Atlantic surface waters (see also our sensitivity experiment assuming 100% in comparison below). We allow biological production under sea ice scaled with the local sea ice thickness.

**4 Control run, scenario experiments, and sensitivity experiments**

The model was spun-up re-starting from an earlier integration and computed into quasi-equilibrium including the sediment over 40,000 years. The equilibrium coefficient $k_d$ was determined through a visual fit to observations of dissolved $^{230}$Th taken from the literature (Bacon and Anderson, 1982; Bacon et al., 1989; Chase et al., 2002; Cochran et al., 1995; Cochran et al., 1987; Colley et al., 1995; Guo et al., 1995; Huh and Beasley, 1987; Moore, 1981; Moran et al., 1997; Moran et al., 1995; Nozaki and Horibe, 1983; Nozaki and Yang, 1987; Nozaki et al., 1987; RoyBarman et al., 1996; Scholten et al., 1995; Vanderloeff and Berger, 1993; Vogler et al., 1998) and combined with the data as given in the GEOTRACES Intermediate Data Product Version 3 (Mawji et al. , 2015; Hayes et al., 2015a,b; Deng et al., 2014). Important global bulk numbers and the parameters for the partitioning coefficient $k_d$ are listed in Table 1.

Meridional sections of dissolved $^{230}$Th concentrations for the Atlantic and Pacific Oceans are given in Figure 3. The vertical distribution clearly shows the increase of concentrations with depth due to the downward transfer of $^{230}$Th with the marine particle flux. A comparison of simulated and observed dissolved $^{230}$Th values in the water column along a meridional Atlantic cross section (Figure 4) indicates that the model dissolved $^{230}$Th is in fairly good agreement with observations, except for a strong overestimation in some bottom water locations. This could

be due to deficiencies in the flow field and corresponding problems or due to the lack of explicit simulations of hydrothermal vents and sediment resuspension. We carried out a respective sensitivity experiment (see discussion further below). A scatter plot of modelled and observed $^{230}$Th values and the locations of stations with observed data are shown in Figure 5. The standard-run export production rates for POC and $CaCO_3$ are given as maps in Figure 6.

An overview concerning the model all model simulations is given in Table 2. We carried out model projections under four different scenarios for changes in calcification (including control runs with constant calcification but rising atmospheric $CO_2$) and four different scenarios of the future development of the atmospheric $CO_2$ concentration (C8.5-2.6, L8.5-2.6, M8.5-2.6, and E8.5-2.6, see Table 2). All scenarios were restarted from the same previously performed spin-up model run reflecting a preindustrial biogeochemical state of the ocean. Throughout the experiments, the ocean circulation field was not changed. The model was integrated during the calendar years 1200-2300 spanning a period of 1100 years. In the model spin-up simulation, the atmospheric $CO_2$ concentration was a prognostic variable. For the computations from 1700 onwards, we prescribed the atmospheric $CO_2$ concentration according to the Representative Concentration Pathways (RCPs) including their extension to year 2300 (van Vuuren et al., 2011) as used in CMIP5 (Coupled Model Intercomparison Project Phase 5). Our simulations follow scenarios RCP 2.6, RCP 4.5, RCP 6.0, and RCP8.5 designed to spanning a range of radiative forcing between 2.6 and 8.5 W/m$^2$ by the year 2100. The respective atmospheric $CO_2$ concentrations are shown in Figure S2. For the decrease in calcification with decreasing pH as well as carbonate saturation, functional relationships between $CaCO_3$ export production and carbonate saturation were chosen which correspond to the respective experiments in Ilyina et al. (2009). For each RCP, we carried out a control simulation with constant calcification, a moderate decrease of $CaCO_3$ production with decreasing saturation, a linear dependency simulation with respectively stronger decrease in $CaCO_3$ production) and an extreme scenario (Figure 2). To date, no clear bulk formulation for the dependency of $CaCO_3$ export on $CaCO_3$ saturation exists. Therefore, the scenarios carried out here are only sensitivity experiments and not overall exhaustive simulations in order to reproduce the entire range of possible changes in ocean biogeochemistry.

As an indication for the robustness of our results, a suite of sensitivity experiments was carried out. In a first set of sensitivity experiments (P1-P5, Table 2), we investigated how much the preindustrial control simulation depends on selected parameter choices. In P1, parameter A for $k_d^{first\ guess}$ was slightly increased from 5.8 to 6.0 in order to test the dependency of the result from small changes in the still not very precisely know $k_d$ value for $^{230}$Th. The effect of release of material from hydrothermal vents and resuspension of material from the sediments, we carried out experiment P2 with a strong increase in scavenging in the lower most wet model grid cells (directly over the ocean floor). To this end, parameter A for $k_d^{first\ guess}$ was increased by 0.5 units in these grid boxes. In our general reference run P0 we applied scavenging to clay, as if the clay flux would only be 25% of its prescribed value. Therefore, we added also run P3, where 100% clay flux was assumed also for the $^{230}$Th scavenging. Finally, in experiments P4 and P5, we switched off the scavenging to POC and $CaCO_3$ respectively in order to see the importance of these two biogenic particle species on the $^{230}$Th distribution.

In a second set of sensitivity experiments, we tested how sensitive the future projection of dissolved $^{230}$Th is in view of other factors than changes in $CaCO_3$ production (S1-S5, see Table 2). Because we use an annually averaged model without seasonal cycle and also employed fixed annul mean velocity field, we have to use approximations in order to see the effect of a change in the velocity field or other changes involving interannual/decadal variability.

5 We use here the time series of the Atlantic Meridional Overturning from the fully fledged Earth system model NorESM for a historic ramp-up and subsequent RCP8.5 forcing (Tjiputra et al., 2016; see Figure 2 therein). In experiment S1, we scaled the three-dimensional velocity field (and also the convective mixing representation) of our simplified model with this time series, leading to an overall reduction in circulation strength. The formulation for this scaling is:

$$Y^{scaled} = (1 + \Delta Overturning(t)) \cdot Y^{reference}$$

with $Y$ being a scalar variable (such as the velocity components in the x-, y-, and, z-directions, the convective adjustment, or other specific biogeochemical parameters), $Y^{scaled}$ the value as updated for the sensitivity experiments, $Y^{reference}$ the original value as used in the reference run, and $\Delta Overturning$ the relative change in Atlantic meridional overturning between a specific year after year 1850 and year 1850 itself ($\Delta Overturning$ would

be zero until 1850). Using this scaling, we also investigated separately from a circulation change the consequences of variations in the maximum nutrient uptake velocity $V_{max}$ in our simple trophic model (a reduction in $V_{max}$ could be seen as a deterioration of growth conditions for plankton under climate change) (run S2), and in the sinking velocity of particles (corresponding to, e.g., a loss in ballast material) (run S3). In experiments S4 and S5 finally, the simultaneous effects of changes in the circulation, $V_{max}$, and the particle sinking velocity were explored for the

moderate (S4) and the extreme (S5) calcification scenarios. The list of potential factors with an influence on $^{230}$Th is much longer, however, our choice of factors covers at least the physical forcing through the velocity field, surface biogeochemistry through $V_{max}$, and three-dimensional biogeochemistry through the particle sinking speed.

## 5 Results and discussion

The results from the sensitivity experiments relative to the preindustrial control run are summarised in Figure 7. Increasing the scavenging equilibrium coefficient in general (run P1, Figure 7a-b) or over the ocean bottom (run P2, Figure 7c-d) corrects to a substantial degree the too high bottom water vales for dissolved $^{230}$Th in the southern Atlantic, but leads to high relative errors in the upper few hundred meters (where $^{230}$Th concentrations are generally low). Too low upper ocean $^{230}$Th concentrations occur also for increasing the scavenging to 100% of the clay flux

(run P3, Figure 7 e-f). Omitting, the scavenging to POC has only minor influence on the $^{230}$Th distribution because of the relatively shallow recycling of POC (run P4, Figure 7g-h). In contrast, cancelling the scavenging to $CaCO_3$ (and only retaining that to POC and clay) leads to a completely unrealistic dissolved $^{230}$Th distribution with at least a fivefold overestimation (run P5, Figure 7i-j; note change of colour code in Figure 7i-j). These results demonstrate that the choice of critical parameters for our preindustrial reference simulation (run P0) results in a reasonable

dissolved $^{230}$Th distribution. The crucial role of $CaCO_3$ particles for determining water column $^{230}$Th has been confirmed.

We discuss now the results for the projections. According to the various scenarios applied, considerable changes in surface alkalinity (not shown) and corresponding changes in $CaCO_3$ export production evolve at high $CO_2$

(Figure 8). The effect is on the average larger in the Pacific than in the Atlantic due to the overall higher modelled

biological production rates of $CaCO_3$ in the Pacific Ocean. The extreme calcification scenario would lead to vastly reduced $CaCO_3$ export production after year 2100 in all RCPs except the most moderate RCP2.6 where even a partial recovery occurs.

For dissolved $^{230}Th$, we show here time series for depth levels 700 m, 2000 m, and 4000 m (Figures S3, 9, and S4, respectively). According to recent intercalibration experiments, still sizable discrepancies exist between the absolute values in $^{230}Th$ measurements from different laboratories, though these measurements show a smaller scatter for replicates within a single laboratory. Intercalibration experiments reveal a standard deviation for measurements on one sample by different laboratories of about 0.07-0.08 dpm/1000l (Anderson et al., 2012). We take 0.075 dpm/1000l as an approximate indicative value for the detection level for changes in dissolved $^{230}Th$ concentrations in seawater in (Figures S3, 9, and S4; see orange lines therein). This may be a somewhat optimistic estimate for samples of past decades, but future developments of measuring techniques could possibly reduce measurement errors and the spread across analyses from different laboratories and application of different measurement methods. The detection level is shown here relative to the modelled preindustrial values. The time of emergence as indicated by the orange line in Figures S3, 9, and S4 is, therefore, the earliest possible time of detection if preindustrial $^{230}Th$ would be known. Figures S3, 9, and S4 show the increase of dissolved $^{230}Th$ activities with depth from 700 m (Figure S3), over 2000 m (Figure 9) to finally 4000 m (Figure S4). At 700 m the increase in $^{230}Th$ activities due to the assumed ocean acidification effect is too small to unambiguously show an effect for RCP2.6 (Figure S3). This picture changes when going down to 2000 m and below, where an effect would be detectable within this century at least for the stronger forcing scenarios RCP8.5 and RCP6.0 (Figures 9 and S4). For constant $CaCO_3$ production, the intermediate and deep-water $^{230}Th$ activities start to rise around year 2100 as well (see black curves in Figures S3, 9, and S4). This effect is due to the increasing dissolution of $CaCO_3$ particles in the water column in parallel with downward mixing of waters that carry anthropogenic loads of dissolved organic carbon and hence subsurface and deep acidification. The effect is most important in areas, where anthropogenic carbon is mixed down quickly and induces a significant shoaling of the $CaCO_3$ saturation level and $CaCO_3$ lysocline as well the Carbonate Compensation Depth through deep-water acidification. Parts of the deep Pacific are not as much influenced by this as compared to the Atlantic within the 21st century. The control run in the figures does not represent the reference run with constant preindustrial atmospheric $pCO_2$ but the run with constant $CaCO_3$ production and rising $pCO_2$ according to the RCP scenarios.

In order to look for most suitable regions for detection of ocean acidification effects on $CaCO_3$ particle fluxes using modern $^{230}Th$ data, we plotted the year of emergence for different reference years and different assumed detection levels (Figure 10). The year of emergence is defined here as the earliest possible year of a potential detection in case that the $^{230}Th$ signal (rate of change over time) is only influenced by a decrease in calcification and not any other processes (such as potential shifts in circulation and associated changes in biological particle production). We choose 2010 as reference year, to which the $^{230}Th$ activity at a later stage would be compared to, and 0.075 dpm/1000l as limit for the detection of a signal. For the moderate calcification scenario, the time of emergence at least at 4000 m would be comparable to the time of emergence as potentially inferable through repeat surface alkalinity measurements (Ilyina et al., 2009) (Figure 10a-b). For the extreme scenario the signal would be identifiable considerably earlier. The signal emerges earliest in deep Pacific waters.

The results on the early warning indicator as shown in Figure 10 are based on the simplifying assumption, that only changes in calcification caused by ocean acidification influence the water column $^{230}$Th concentrations under progressing acidification and climate forcing. We, therefore, also mimicked the effect of interannual/decadal variability in circulation and biogeochemical parameters (run S1-S5, Table 2). Comparing the projections under RCP8.5 and the moderate calcification scenario for the case without interannual/decadal variability (run M8.5, Table 2) with the one including variability (run S1-3, Table 2) for the middle of this century reveals differences in the order of the assumed detection level for changes in the velocity field and the particle sinking velocity and smaller changes for varying $V_{max}$ (Figure 11a-f). The effect on modulating water column $^{230}$Th can be quite variable for different oceanic domains. Deviations can add-up or compensate each other locally. For simultaneous changes of circulation, $V_{max}$, and particle sinking velocity, the overall deviations from the run without variability amount to 2-3 times the theoretical detection limit (Figure 11g-h). We repeated the analysis of the time of emergence with the runs including interannual variability for the moderate (S4) and also the extreme (S5) scenario. For this analysis, we again chose 2010 as the reference year but increased the detection limit to 2.5*0.075 dpm/1000l. Under these more realistic conditions, the time of emergence would be shifted to the end of the century for the moderate calcification scenario and to the middle of the century for the extreme scenario (Figure 12).

The question arises, whether surface alkalinity and deep $^{230}$Th measurements possibly could be combined successfully in order to detect ocean acidification impacts on calcification. We could, so far, not detect any simple relationship between surface changes in $CaCO_3$ export production, surface alkalinity, and $^{230}$Th in spite of the conceptually clear interdependencies. In order to illustrate this, we plotted changes in surface alkalinity and 3000 m $^{230}$Th over time for the run with moderate calcification scenario and RCP8.5 forcing between 2040 and 2010. The distributions show a somewhat consistent picture without interannual/decadal variability (run M8.5, Figure 13a-b) with general rises in both surface alkalinity and deep $^{230}$Th. For the case including interannual/decadal variability (run S4, Figure 13c-d) the temporal gradients in both variables become smaller and the deep $^{230}$Th even may reverse. This does not mean that it would be impossible to construct a combined Alk-$^{230}$Th tracer, only that such a tracer cannot be derived from this study..

As compared to the detection approach for ocean acidification impacts through total alkalinity measurements as pursued by Ilyina et al. (2009), the $^{230}$Th method presented here results in similar time of emergence of a signal reflecting changes in biocalcification if one omits interannual/decadal variability and other possible influence factors. The probably most important limitation of the approach here is the fixed velocity field which does not vary realistically with seasons, interannual variability, climatic variability modes (such as El Niño Southern Oscillation or North Atlantic Oscillation). Also the effect of boundary scavenging (Anderson et al., 1983; Roy-Barman et al., 2009), i.e. the transport of dissolved $^{230}$Th from areas of low particle concentrations to those with high particle concentrations (especially at ocean margins at shelf seas) is not spatially resolved in our model. Further uncertainties are associated with the choice of the $k_d$ values and particle specific reactivity.

The extreme scenario on pH-dependent $CaCO_3$ production is likely to be an overestimation. According to the results from this study and Ilyina et al. (2009) a respective large change in the real world would probably have been detected through the alkalinity signal and other methods such as remote sensing and sediment trap measurements already. In addition, the reaction of $CaCO_3$ shell producing organisms to sinking carbonate

saturation levels is not a simple function and can vary between taxa (Kroeker et al., 2013).

The detection method for ocean acidification impacts through $^{230}Th$ would work best, if the preindustrial values for dissolved oceanic $^{230}Th$ activities could reliably be reconstructed. From core top samples and corals, one could possibly determine whether the particle attached $^{230}Th$ activities and hence those of the dissolved fraction would

have undergone any variations over the last centuries and millennia. However, the overall uncertainties associated with the formation and analysis of the paleo-record values may be too high to provide an accurate enough baseline for comparisons with modern water column measurements.

**6 Conclusions**

In this study, we investigated the potential of the particle reactive radionuclide $^{230}Th$ for detection of reduced calcification by biota due to progressing ocean acidification. The time of emergence of a dissolved $^{230}Th$ activity signal with respect to a reference year and reference activity is about the same as for detection approaches employing alkalinity if one disregards effects of interannual/decadal variability in the flow field and other influence factors than ocean acidification induced changes in $CaCO_3$ export production. Taking into account

interannual/decadal variability including ocean circulation changes may delay the signal emergence by several decades. Nevertheless, regular selected reoccupations of a series of deep stations in the Pacific Ocean and Southern Ocean with highest quality $^{230}Th$ measurements would be helpful to accompany alkalinity measurements that are easier to do in order to see whether both tracers give consistent results. Surface alkalinity measurements include signals of natural variability on seasonal and multiyear scales. Likewise, changes in ocean circulation and changes

in biological particle production due to other processes than ocean acidification may lead to changes in the marine $^{230}Th$ distribution (and alkalinity). As the deep ocean is less prone to effects of natural variability and the quality of observations does not change with depth, deep ocean observations of $^{230}Th$ could be advantageous for monitoring and detecting ocean acidification effects on calcification. In any case, $^{230}Th$ represents a fascinating magnifying glass for changes in ocean surface processes seen through the deep ocean signal. Its potential has not

yet been fully exploited.

**Acknowledgment:** This work was partially supported through the "European Project on Ocean Acidification" EPOCA and project "Changes in Carbon Uptake and Emissions by Oceans in a Changing Climate" CARBOCHANGE that received funding from the European Community's Seventh Framework Programme (FP7)

under grant agreements no. 211384 and no. 264879 respectively. The Research Council of Norway supported this study through project "Overturning circulation and its implications for global carbon cycle in coupled models" (ORGANIC; grant no. 239965) and the nationally coordinated project "Earth system modelling of climate variations in the Anthropocene" (EVA; grant no. 229771). This is a contribution to the Bjerknes Centre for Climate Research (Bergen, Norway). A part of the computations were carried out under project NN2980K at the Norwegian

Metacenter for Computational Science (NOTUR) and its dedicated storage and archiving project NorStore (NS2980k). This is an in-kind contribution to the GEOTRACES project. We would like to thank the GEOTRACES scientists for the interesting discussions during several workshops and for the fabulous new observational data.

Jerry Tjiputra kindly made available the AMOC data from NorESM. Two constructive referees helped to improve this manuscript.

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

# Tables:

| Key variable as simulated | unit | Control run value |
|---|---|---|
| Atmospheric $CO_2$ mixing ratio | [ppm] | 277.6 |
| POC export production (pre-industrial) | [Gt C yr$^{-1}$] | 9.25 |
| $CaCO_3$ export production (pre-industrial) | [Gt C yr$^{-1}$] | 1.07 |
| Coefficient A for $k_d^{first\ guess}$ | - | 5.8 |
| Coefficient B for $k_d^{first\ guess}$ | - | 0.4 |
| Coefficient $C_{CaCO3}$ | - | 3.0 |
| Coefficient $C_{POC}$ | - | 0.3 |
| Coefficient $C_{clay}$ | - | 2.3 |

**Table 1: Summary of control run results (run P0, see Table2).**

| Name | $CO_2$-scenario | Calcification Scenario | Sensitivity experiment description |
|---|---|---|---|
| P0 | Preindustrial | Constant | Pre-industrial control run, all variables in quasi-equilibrium |
| P1 | Preindustrial | Constant | Coefficient A for $k_d^{first\ guess}$ increased from 5.8 to 6.0 |
| P2 | Preindustrial | Constant | Coefficient A for $k_d^{first\ guess}$ increased by 0.5 in bottom grid cells |
| P3 | Preindustrial | Constant | Clay flux increased from 25% to 100% dust input |
| P4 | Preindustrial | Constant | No scavenging to POC |
| P5 | Preindustrial | Constant | No scavenging to $CaCO_3$ |
| C8.5 | RCP8.5 | Constant | RCP8.5 control, only change is atmospheric $CO_2$ |
| C6.0 | RCP6.0 | Constant | RCP6.0 control, only change is atmospheric $CO_2$ |
| C4.5 | RCP4.5 | Constant | RCP4.5 control, only change is atmospheric $CO_2$ |
| C2.6 | RCP2.6 | Constant | RCP2.6 control, only change is atmospheric $CO_2$ |
| L8.5 | RCP8.5 | Linear | Calcification reduction |
| L6.0 | RCP6.0 | Linear | Calcification reduction |
| L4.5 | RCP4.5 | Linear | Calcification reduction |
| L2.6 | RCP2.6 | Linear | Calcification reduction |
| M8.5 | RCP8.5 | Moderate | Calcification reduction |
| M6.0 | RCP6.0 | Moderate | Calcification reduction |
| M4.5 | RCP4.5 | Moderate | Calcification reduction |
| M2.6 | RCP2.6 | Moderate | Calcification reduction |
| E8.5 | RCP8.5 | Extreme | Calcification reduction |
| E6.0 | RCP6.0 | Extreme | Calcification reduction |
| E4.5 | RCP4.5 | Extreme | Calcification reduction |
| E2.6 | RCP2.6 | Extreme | Calcification reduction |
| S1 | RCP8.5 | Moderate | As M8.5, but overturning reduced with rising $CO_2$ |
| S2 | RCP8.5 | Moderate | As M8.5, but $V_{max}$ for nutrient uptake reduced with rising $CO_2$ |
| S3 | RCP8.5 | Moderate | As M8.5, but particle sinking velocity reduced with rising $CO_2$ |
| S4 | RCP8.5 | Moderate | As M8.5, but changes as in S1M8.5, S2M8.5, S3M8.5 together |
| S5 | RCP8.5 | Extreme | As E8.5, but changes as in S1M8.5, S2M8.5, S3M8.5 together |

**Table 2: Model runs overview.**

# Figures:

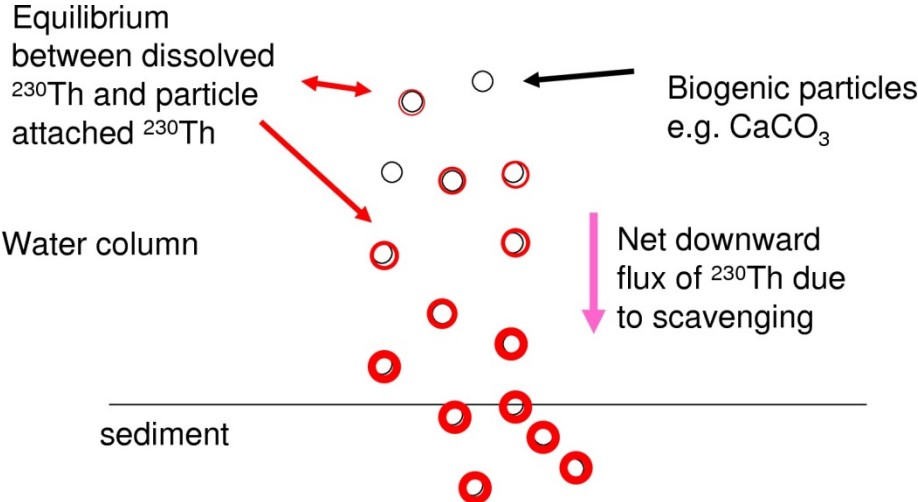

**Figure 1:** Schematic illustration of the equilibration process between the dissolved and particle-attached phases of $^{230}$Th and the increasing concentrations downward in the water column.

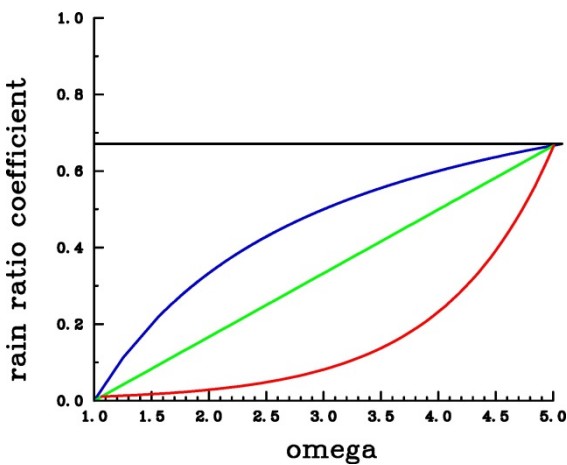

**Figure 2:** Assumed sensitivities of CaCO$_3$ export production in response to changes in calcite saturation. The rain ratio coefficient is used as factor "a" applied to the right hand side of the equation for CaCO$_3$ production (see eq. 1). For the control run, "a" was fixed at each grid point according to the prevailing pre-industrial carbonate saturation (following the moderate sensitivity). Blue: Moderate sensitivity. Green: Linear sensitivity. Red: Extreme sensitivity. The shape of the curves would 
15    look similar at each grid point; only the pre-industrial cross-over point would be different.

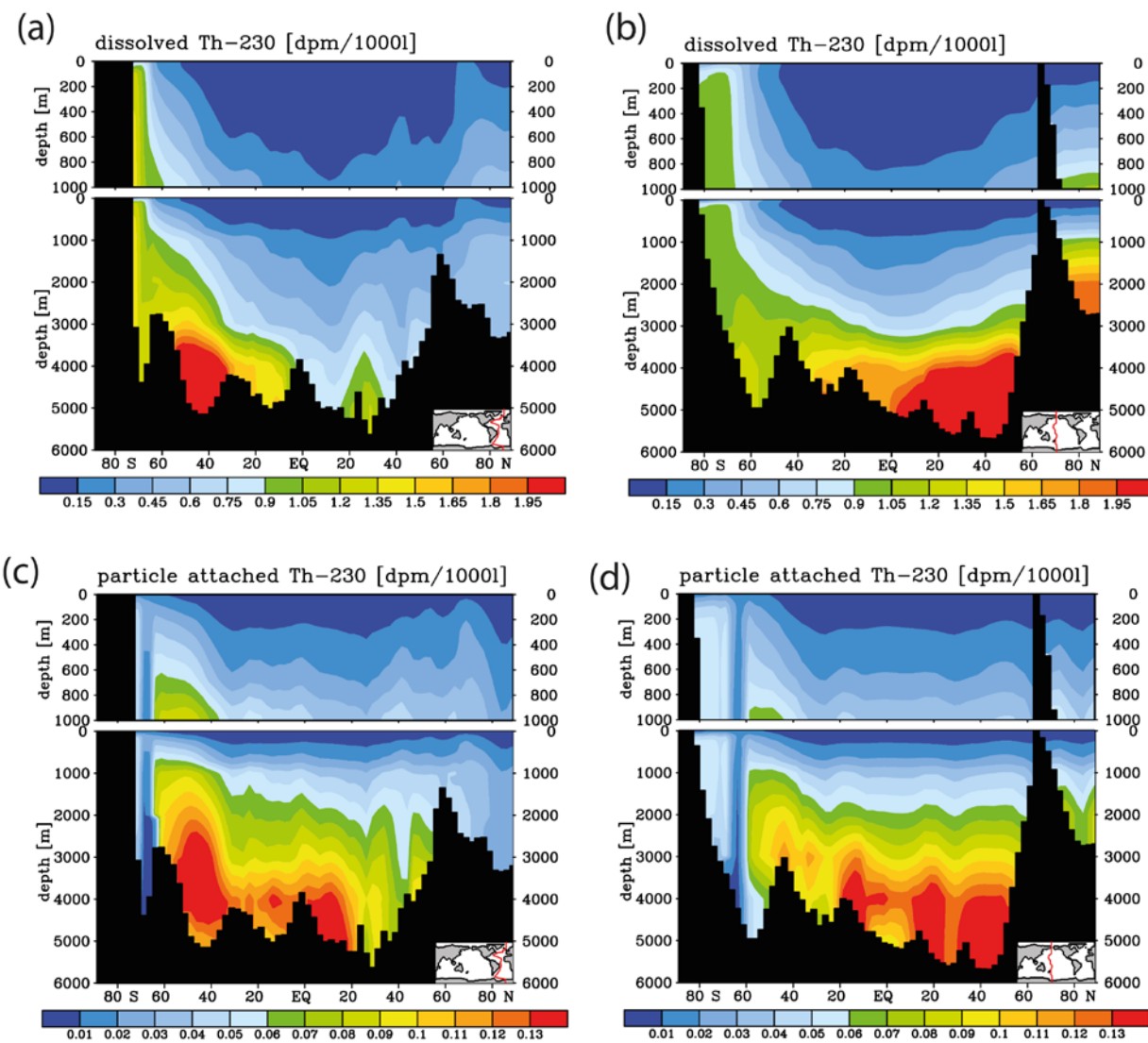

**Figure 3:** Meridional $^{230}$Th cross sections in [dpm/1000l] (dpm = disintegration per minute) for the model control run under pre-industrial atmospheric $CO_2$. (a) Dissolved, Atlantic. (b) Dissolved, Pacific. (c) Particle attached, Atlantic. (d) Particle attached, Pacific.

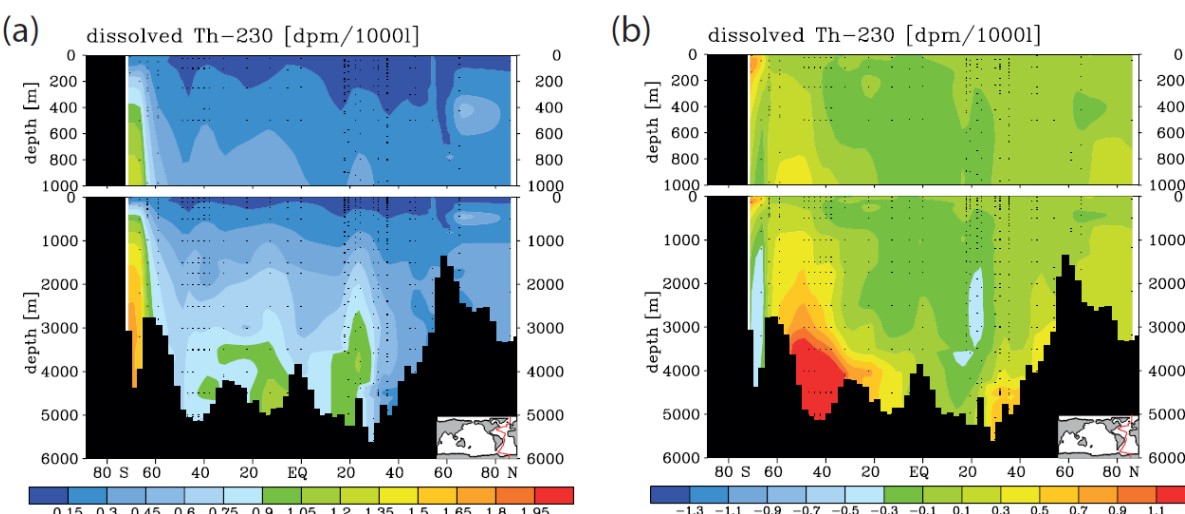

**Figure 4:** Meridional dissolved $^{230}$Th Atlantic Ocean cross section in [dpm/1000l] (dpm = disintegration per minute): (a) Interpolation from observations (data from the GEOTRACES intermediate data product; for references, please see text). (b) Difference between the model values from the control run without $CO_2$ emissions and the observations.

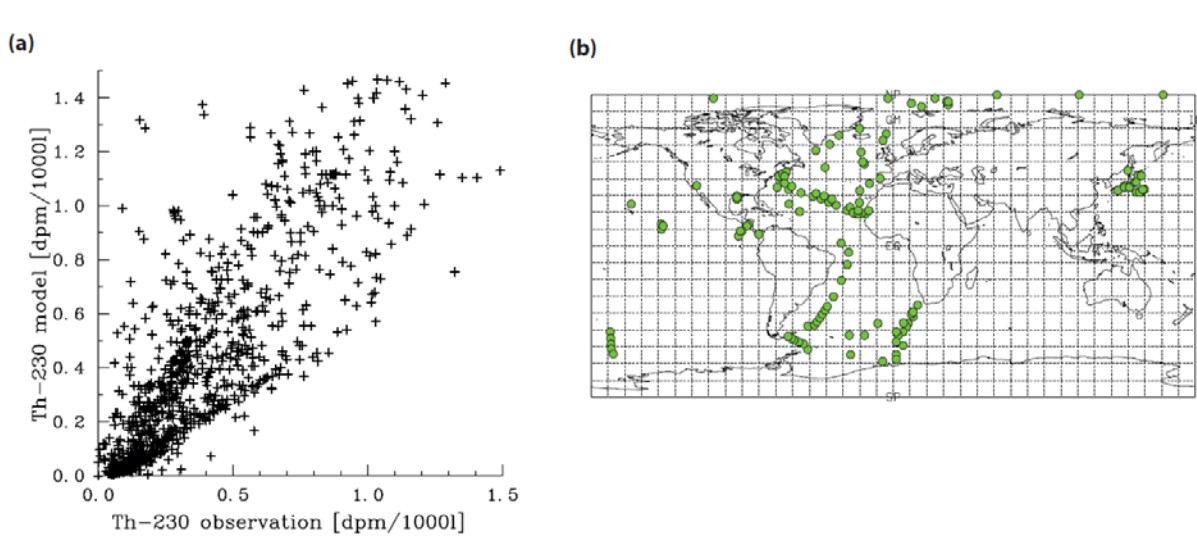

**Figure 5:** Comparison of dissolved $^{230}$Th model results with observations (for references of observations see text). (a) Correlation between observations and modelled values. (b) Location of all vertical profiles as used in this work.

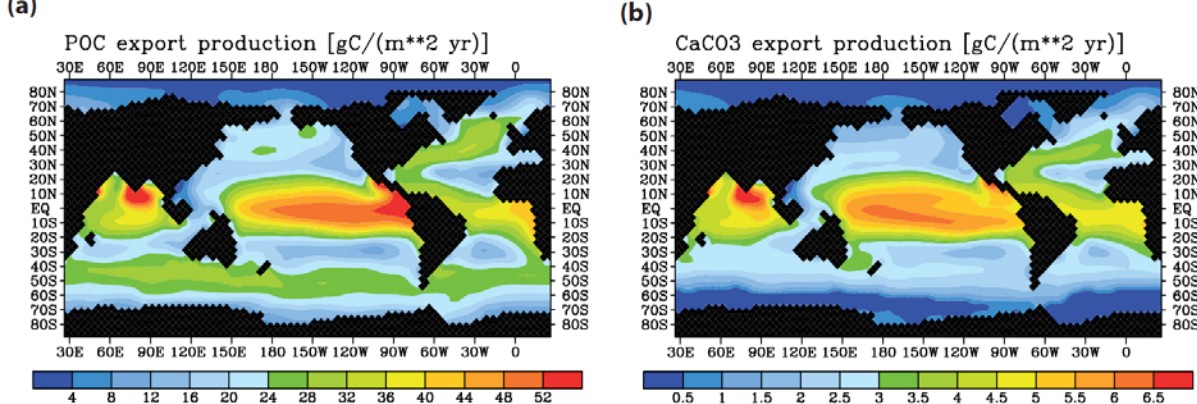

**Figure 6:** Model control run results for (a) biological export production of particulate organic carbon and (b) CaCO₃ export.

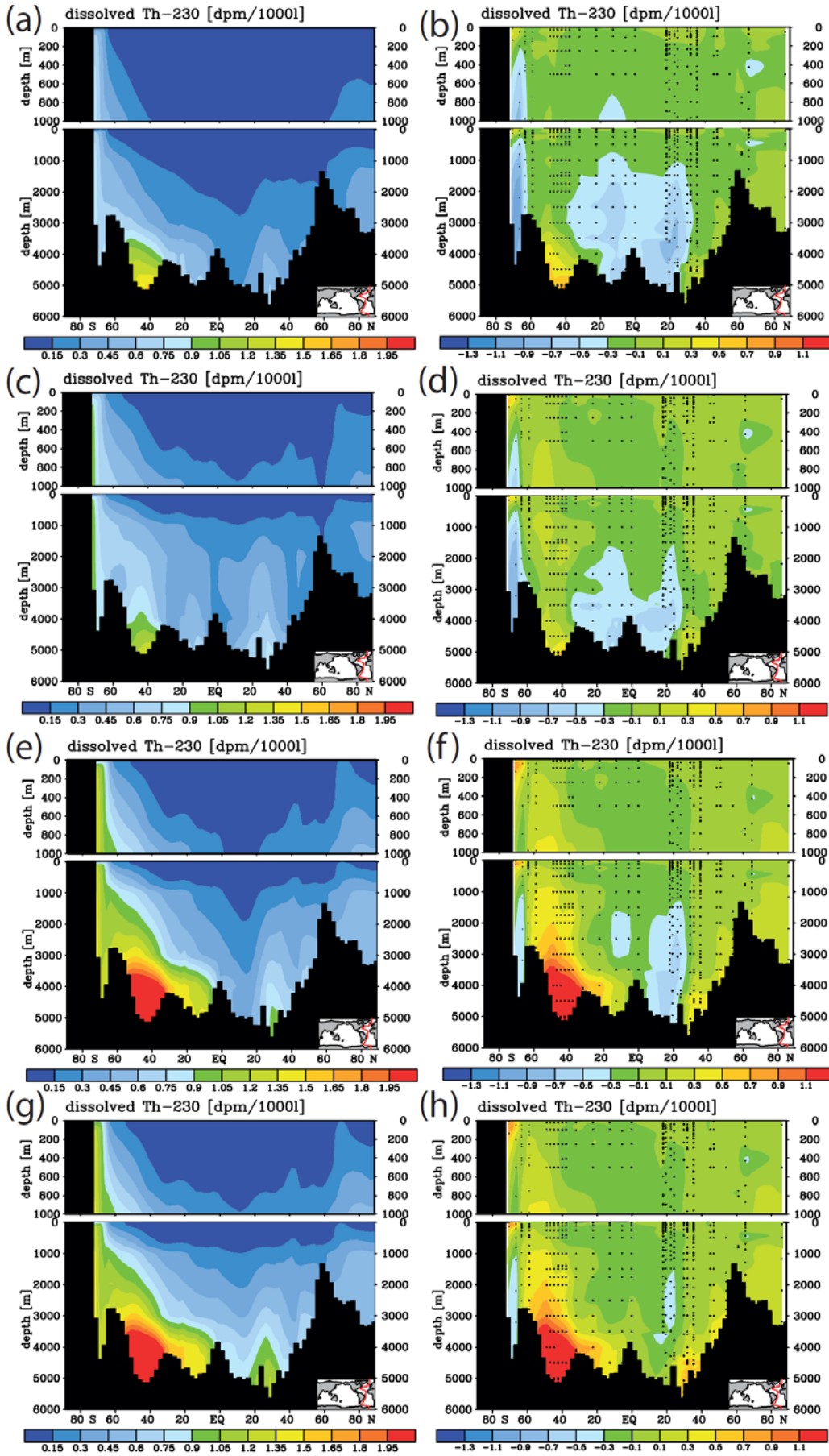

**Figure 7 (a)-(h):** Meridional dissolved $^{230}$Th Atlantic Ocean cross sections in [dpm/1000l] (dpm = disintegration per minute) for sensitivity experiments. (a) $K_d$ increased everywhere (by log($K_d$)=+0.2). (b) Same as a, but difference model minus observations. (c) $K_d$ increased only in the model grid cells directly over the ocean floor (by xyz). (d) Same as c, but difference model minus observations. (e) Clay flux increased to 100% of control run value. (f) Same as e, but difference model minus observations. (g) No scavenging to POC. (h) Same as g, but difference model minus observations. (Observed data from the GEOTRACES intermediate data product; for references, please see text.)

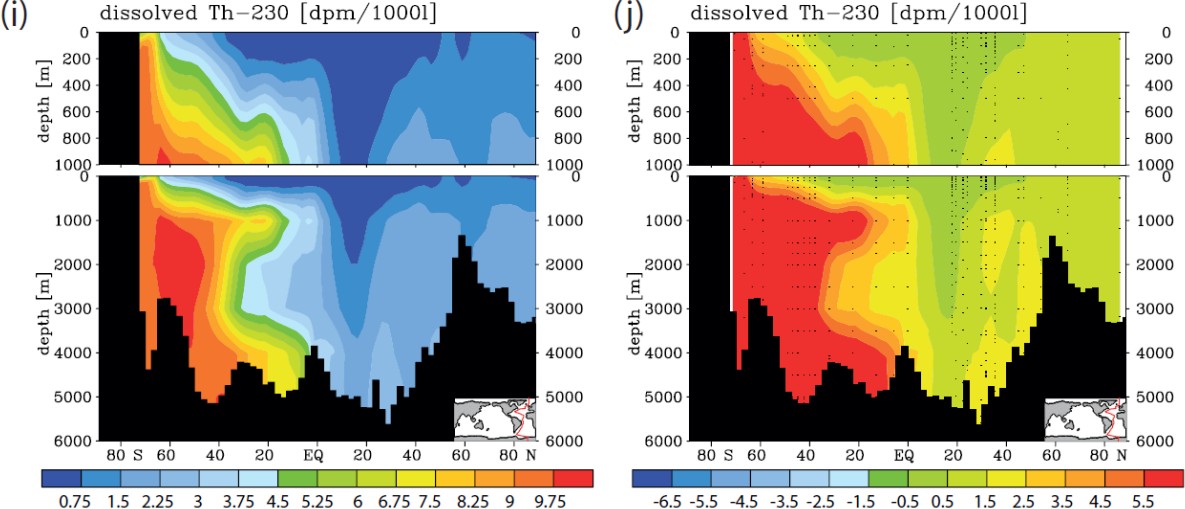

**Figure 7 continued (i)-(j):** (i) No scavenging to CaCO$_3$. (j) Same as i, but difference model minus observations. Please, note changes in colour codes relative to Figure 7 (a)-(h). (Observed data from the GEOTRACES intermediate data product; for references, please see text.)

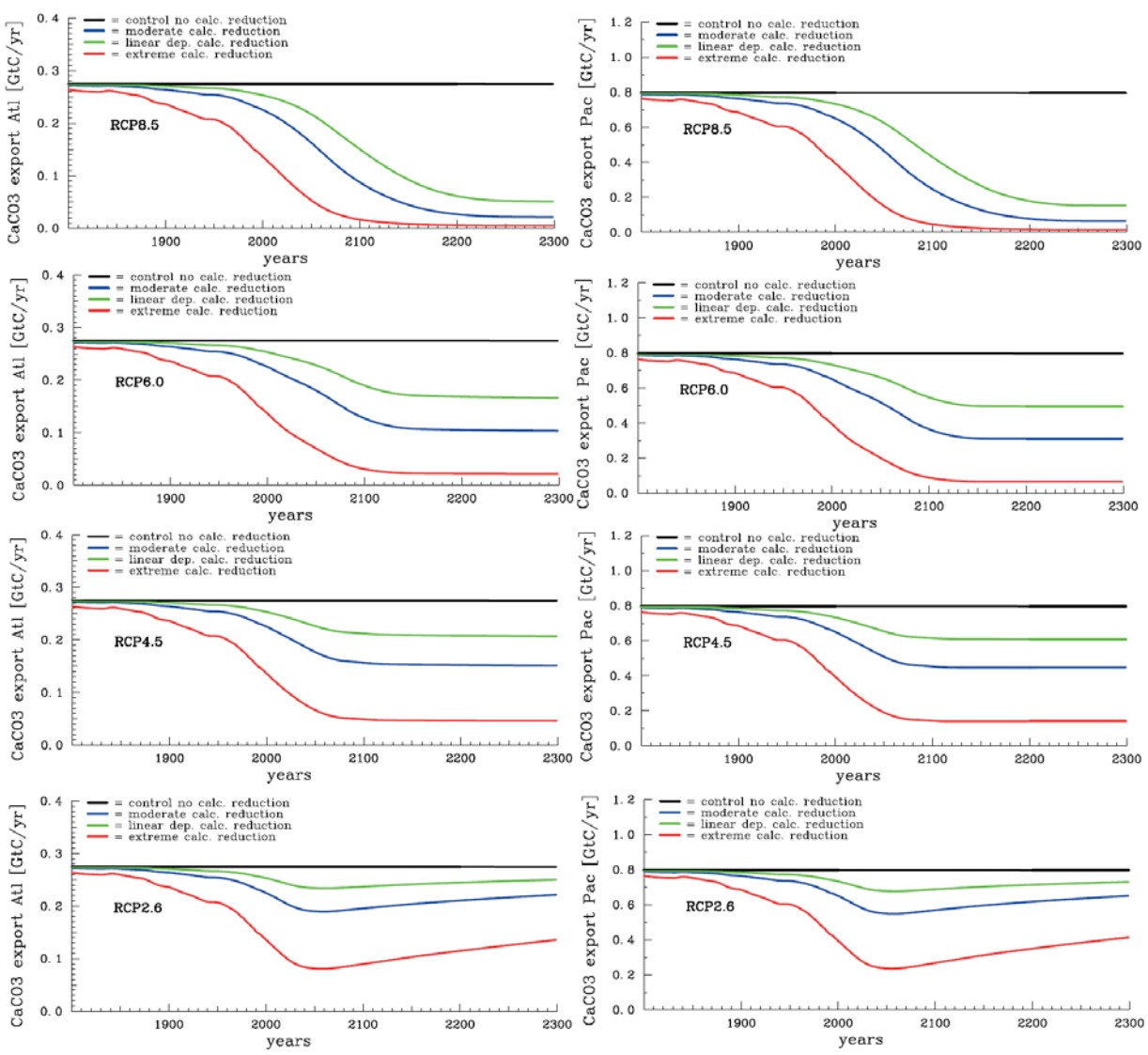

**Figure 8:** Temporal evolution of Atlantic (left column) and Pacific (right column) annual mean CaCO$_3$ export production under the different scenarios for the sensitivity of calcification under high CO$_2$ (unit: GtC yr $^{-1}$). From top to bottom for greenhouse gas scenarios RCP8.5. RCP 6.0, RCP4.5, and RCP2.6.

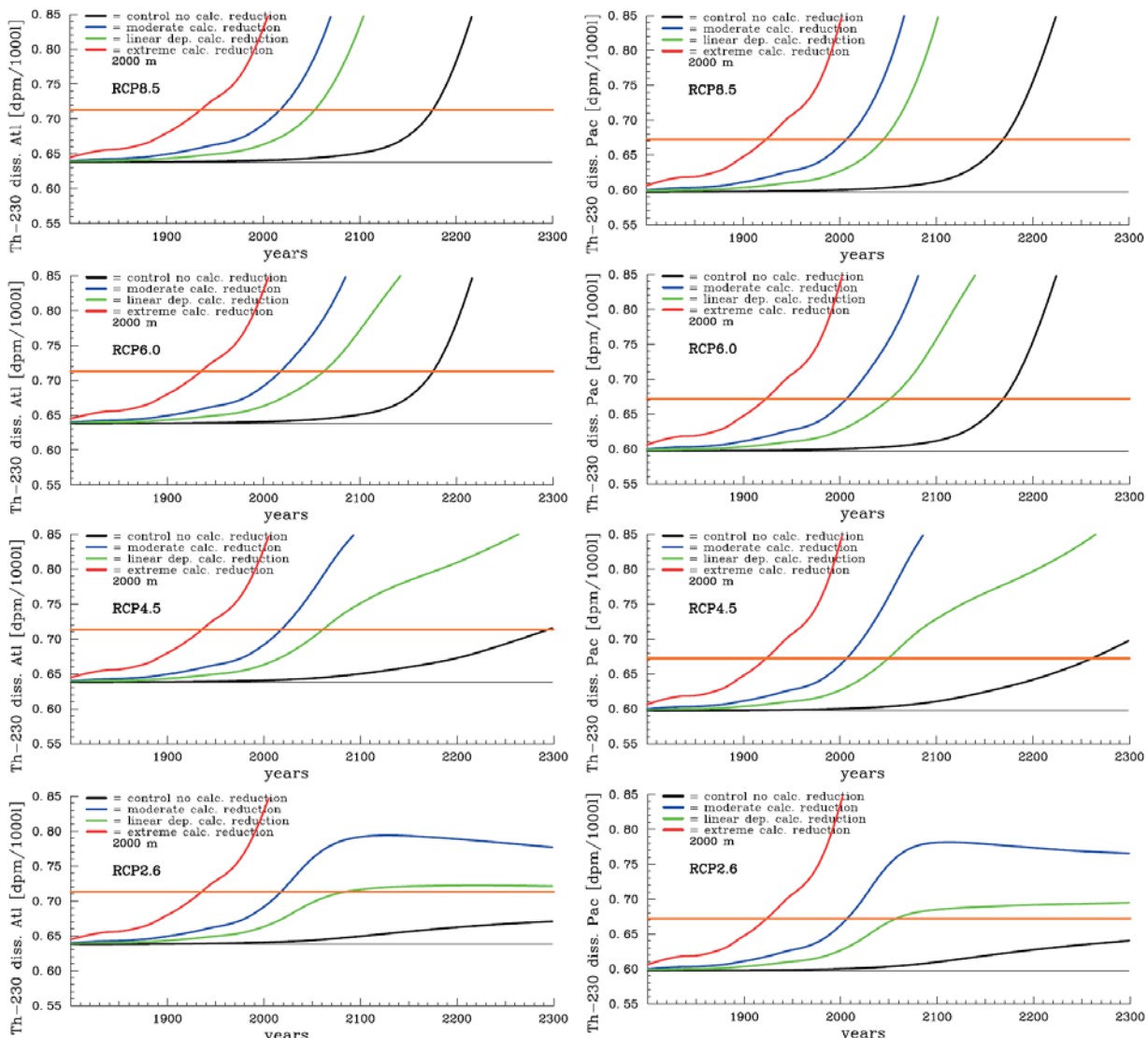

**Figure 9:** Time series for the evolution of mean Atlantic (left column) and Pacific (right column) dissolved $^{230}$Th concentrations at 2000 m under the different scenarios for reduction of calcification under high $CO_2$ (unit: dpm/1000l). From top to bottom for greenhouse gas scenarios RCP8.5. RCP 6.0, RCP4.5, and RCP2.6. The orange line indicates the theoretical detection limit for changes with respect with respect to the pre-industrial. (Respective diagrams for depth levels 700 m and 4000 m are given in Figures S3 and S4.)

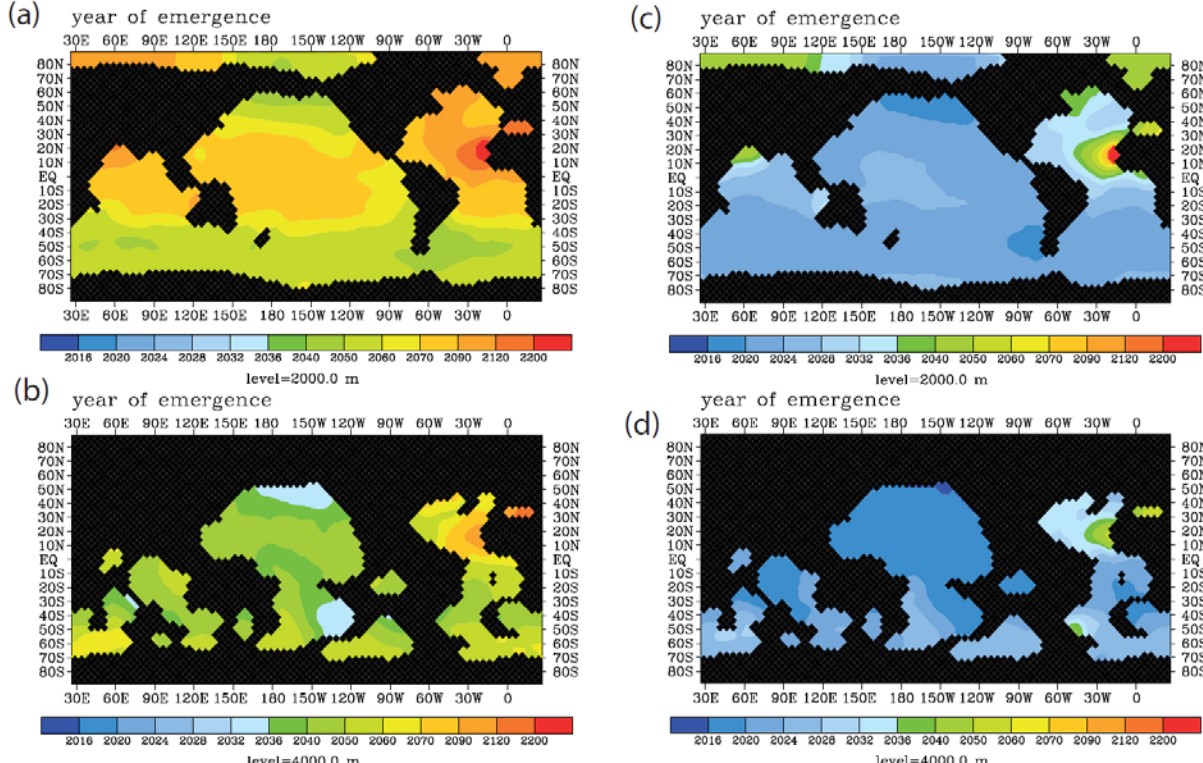

**Figure 10:** Prospective year of emergence for an ocean acidification induced signal in dissolved [230]Th activity as derived from the model runs. Shown are the calendar years of emergence for the depth levels 2000m and 4000m. All figures are shown for the strong RCP8.5 scenario concerning atmospheric $CO_2$ concentration and the moderate scenario of calcification decrease with saturation state, year 2010 as reference year for [230]Th activity and 0.075 dpm/1000l as analytical threshold between different samples. (a) For the moderate calcification scenario, at 2000 m depth. (b) Same as (a) but for depth level 4000 m. (c) For the extreme calcification scenario, at 2000 m depth. (d) Same as (c) but for depth level 4000 m.

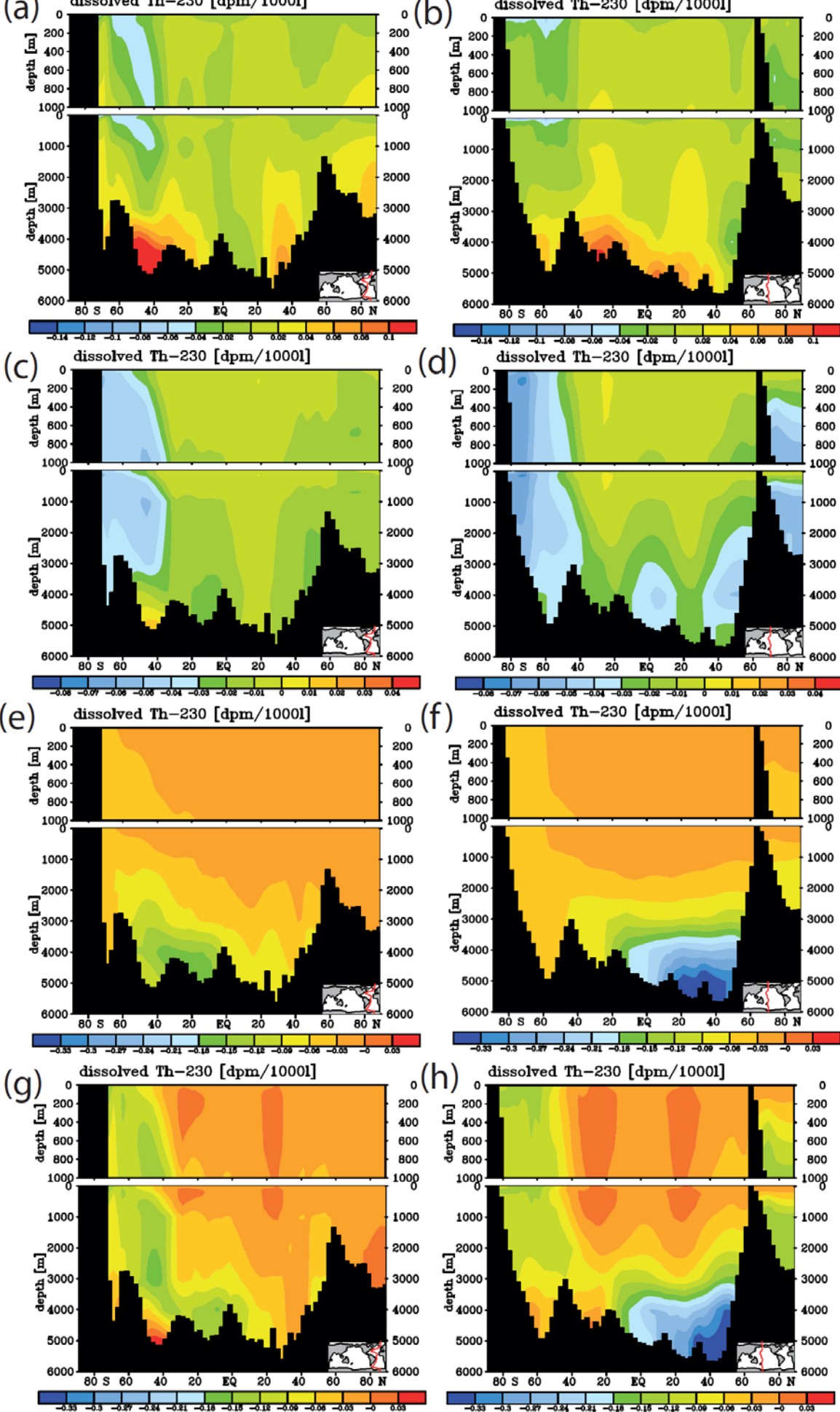

**Figure 11:** Meridional cross sections for sensitivity experiments concerning the projections including a representation of interannual/decadal variability. Shown are differences in dissolved $^{230}$Th for forcing relative to the normal run with RCP8.5 and the moderate calcification scenario for year 2050. (a) Change in velocity field (for details see text). (b) As (a) but for the Pacific. (c) Change of $V_{max}$ for phosphate uptake (for details see text), Atlantic. (d) As (c) but for the Pacific. (e) Change of the sinking velocity of particulate matter (for details see text). (f) As (e) but for the Pacific. (g) Simultaneous change of velocity field, $V_{max}$, and sinking velocity.

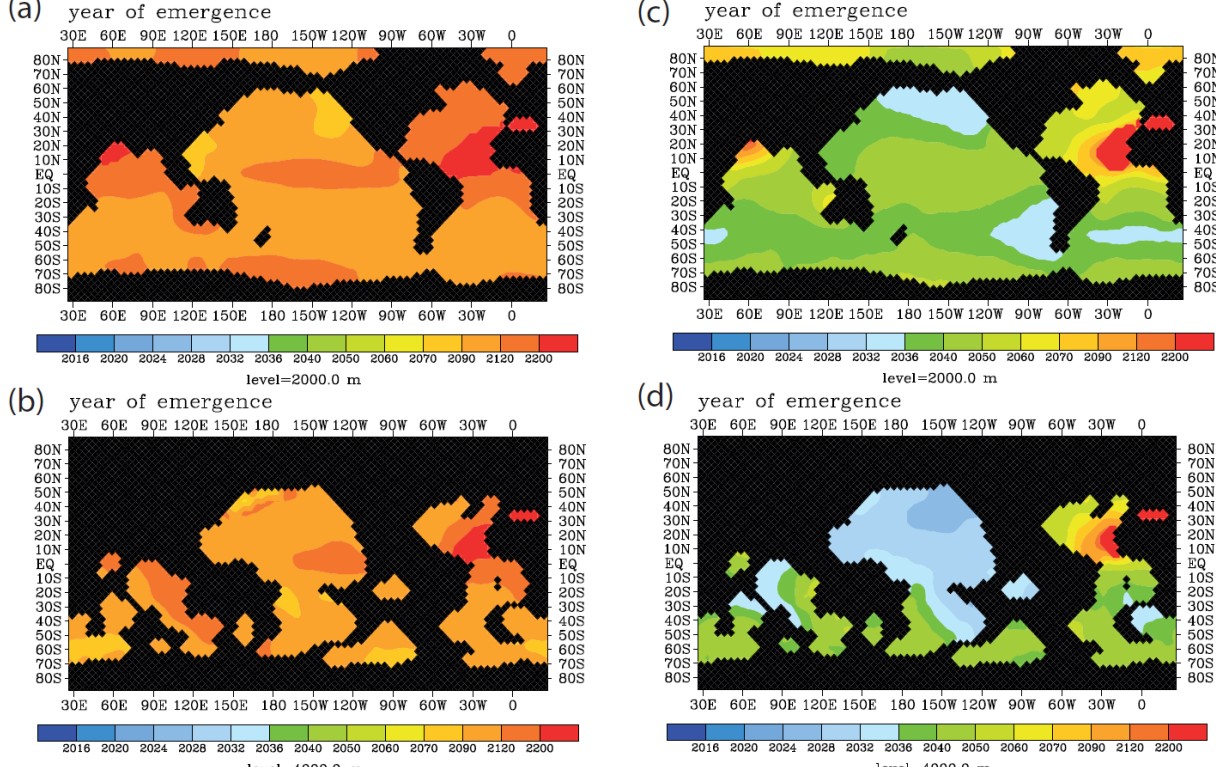

**Figure 12:** Prospective year of emergence for an ocean acidification induced signal in dissolved $^{230}$Th activity as derived from the model runs. Shown are the calendar years of emergence for the depth levels 2000m and 4000m. All figures are shown for the strong RCP8.5 scenario concerning atmospheric $CO_2$ concentration and the moderate scenario of calcification decrease with saturation state, year 2010 as reference year for $^{230}$Th activity and 2.5·0.075 dpm/1000l as analytical threshold between different samples. In contrast to Figure 11, here an analysis including interannual/decadal variability in selected biogeochemical parameters is shown. (a) For the moderate calcification scenario, at 2000 m depth. (b) Same as (a) but for depth level 4000 m. (c) For the extreme calcification scenario, at 2000 m depth. (d) Same as (c) but for depth level 4000m.

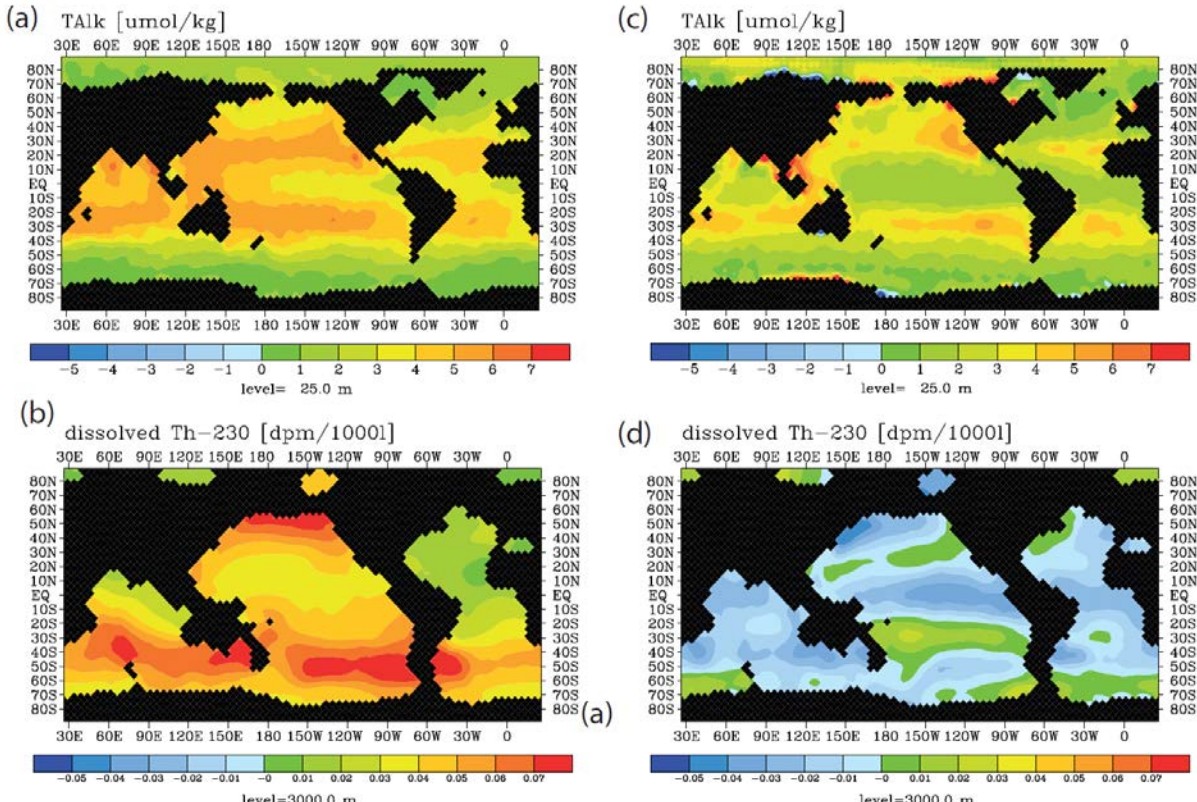

**Figure 13:** Surface alkalinity changes and deep ²³⁰Th changes during 2040-2010 as for the RCP8.5 projection using moderate calcification changes. Left column without variability, (a) at the sea surface, (b) at 3000 m. Right column with decadal parameter variability, (c) at the sea surface, (b) at 3000 m.