# Peer review of "The potential of 230Th for detection of ocean acidification impacts on pelagic carbonate production"

_Biogeosciences, 2017_

## Referee Comment (RC1) · Anonymous Referee #1 · 24 Oct 2017

This modelling study seeks to test whether the oceanic distribution of dissolved 230Th could serve as an indicator of reduced biogenic CaCO3 formation as the ocean acidifies due to anthropogenic CO2 emissions. It proposes that 230Th concentrations, particularly in the deep ocean, may be a more sensitive indicator of such change than direct observations of changing alkalinity in the surface ocean.

In some ways, the modelling work described is a rather incremental advance relative to earlier work assessing the 230Th response to change in the CaCO3:POC ratio presented in Heinze et al. 2006. The present manuscript, however, focuses specifically on testing how this ratio might be influenced by future ocean acidification, and whether

this could be detected by 230Th measurements.

The ability to detect systematic change in the production of biogenic CaCO3 in response to ocean acidification would be a useful tool, making this modelling endeavour potentially useful. The idea that 230Th may allow such detection is not intuitive, but interesting and worthy of consideration. So the general direction of this contribution is welcome. I am, however, unsure from the present manuscript that the utility of 230Th to assess CaCO3 flux has been demonstrated.

1. Threshold for detection The authors assume that detection of change in 230Th depends only on the analytical uncertainty of measurement. Measurements of 230Th in seawater shows significant spatial and temporal variation, however, which far exceed measurement uncertainty. Some of this variation reflects known processes, such as productivity or large-scale circulation, which cause consistent spatial patterns. But other variation is akin to 'noise', caused by seasonality of particle flux, eddying circulation, variability in boundary scavenging etc. To assess the possibility to detect change in the profile of 230Th due to variation in the composition of settling particles requires consideration of the natural variability of the 230Th field. One way to consider this might be to statistically compare closely spaced samples in the ever-growing observational 230Th dataset to assess small-scale natural variability. My guess is that a more realistic detection threshold is likely to be 2 to 3 times higher than the value assumed in this study. That would not prevent detection in the deep ocean (e.g. in Fig 9) but would delay the date of detectability in that setting, and would prevent detection at shallower depths. Intuitively that seems realistic given that intermediate depths typically show quite large (and presently poorly explained) temporal changes in 230Th concentrations.

2. Sensitivity to other changes To be a useful monitor for CaCO3 flux change, future 230Th concentrations must be more responsive to that process than to other possible changes. There is very little consideration in the manuscript of other likely controls on the 230Th distribution. These might include future changes in circulation driven by

changing wind fields or freshwater inputs; changing productivity of organic carbon due to circulation changes; changing remineralisation of organic carbon due to altered O2 levels; changing fluxes of silicate dust due to changing winds and continental aridity; or changing ballasting related to ecosystem change. I do not have an instinct about whether any such changes are likely to generate substantial changes in the 230Th field, but this seems a fundamental issue for a modelling study such as this one to address. Can the authors do more to assess whether CaCO3 fluxes are really the dominant control on 230Th change? Or only one of several global changes that will alter the field?

3. Accuracy of the model The 230Th model used is well established and has been thoroughly documented in the literature before, but there are some presentational issues in the present manuscript that limit the reader's ability to assess its prediction of future 230Th change:

i.          Since     Heinze     2006,     there     are     significantly     more 230Th     observations,     including     long     ocean     sections     (see http://www.egeotraces.org/?group=Dissolved%20Natural%20Radionuclides,variable=Th%20230%20dissolved). It is now possible to directly compare modelled sections (e.g. Fig 3) with observations, and this should be done in this manuscript. Doing so reveals some quite important discrepancies, particularly in the deep ocean which is being touted here as a sensitive indicator for changing CaCO3 fluxes. These discrepancies include deviations related to scavenging at the seafloor and in MOR plumes. If these processes are not considered, the deep-ocean sensitivity of 230Th to downward particle flux may well be overestimated.

ii. Perhaps I have misunderstood, but Figures 8-10 indicate that even the control run shows a significant change in deep 230Th, despite the lack of CaCO3 change in this run. This is puzzling, and seems to suggest a problem with the long-term handling of 230Th in the model?

iii. Less significant, but it would also be good to see how the model predicts change as a profile or section, rather than as a timeseries at a single depth. As you go to greater depth in the ocean, the residence time of 230Th increases, so change might be slower, but the flux of organic carbon decreases so the influence of a CaCO3 change will be more important. Seeing how such depth-related effects compete in the model would be interesting, and help to assess how realistic it is in representing the natural cycle.

Overall, these three concerns leave me unconvinced that this study is ready for publication. The idea of using 230Th to assess CaCO3 fluxes is interesting, however, so I'd encourage the authors to seek to address these issues. A revised version of the work could then be a useful contribution.

Other comments:

P4-6: The description of the model set up could be reduced, given that this is a previously described model, and that some aspects (e.g. C isotopes) are not relevant to this study.

P7: It's good to see the GEOTRACES data used, but the source of this data is strangely attributed. Neither de Baar nor Boyle were involved in collection of 230Th data. Please cite the relevant papers directly for this data (e.g. Hayes et al., Deng et al) in addition to the Mawji et al. paper.

Are any spatial changes expected because of changing rain-ratio? High latitude waters will decrease saturation faster than mid-latitude, so changes may be more acute there. I wonder if looking at the relative change in 230Th between regions may be a more sensitive indicator of the specific response to changing carbonate saturation than the general deep-ocean response?

―――――――――――――――――――――――――

---

## Referee Comment (RC2) · Anonymous Referee #2 · 26 Oct 2017

In this manuscript Heinze et al. explore the feasibility of using measurements of dissolved 230Th in the deep ocean to detect changes in calcification as a result of ocean acidification. The idea being it would be nice to be able to "check" the response of carbonate export/calcification to ocean acidification with observations of changes in both parameters. They argue that this approach is complementary to using measurements of alkalinity in the surface ocean to detect changes in calcification. This is an intriguing idea, especially the idea that surface changes are 'magnified' in the deep water through 230Th. And I fully agree with the authors' concluding sentence that the full potential of 230Th has yet to be realised. However, I have some reservations about this manuscript.

[Figure]

First, the idea is not actually new, having been proposed over 10 years ago by the first author (Heinze et al, 2006).

More importantly, and particularly as the current work can be considered a 'follow up' on the initial work, this work is missing a key sensitivity analysis.

Stepping back, the authors have shown that it's quite possible that changes in carbonate production and export will result in measureable changes in 230Th concentrations in the deep sea in coming decades or at least centuries. However, they have not adequately demonstrated that it will be possible to distinguish between the different 'parameterisations', or 'sensitivities' of calcification to OA, using the 230Th measurements, which is the premise of this work. In their experiments, where only CO2 and calcification change, yes there are clear differences in the time of emergence for the different parameterisations. However, in the real world there will almost certainly be changes in POC, CaCO3 and opal that are independent of OA, driven by changes in stratification, temperature, dust deposition, even anthropogenic nutrient input. Do the different Th trajectories and times of emergence stay distinctly different when you add in these other changes? Furthermore, ocean circulation will likely change and that will change 230Th distributions independent of changes in particle flux and composition. Again, can you distinguish between different OA sensitivities once you factor in a range of ocean circulations? It all comes down to the sensitivity of 230Th distributions to OA compared to their sensitivity to other perturbations.

The authors mention that changes in deep ocean 230Th concentration emerge at about the same time as changes in alkalinity, and point out that the two tracers offer complementary information. I think this is an excellent point. My feeling is a study that combines these two tracers, together with a sensitivity analysis as suggested here, could potentially be quite useful in showing how the two tracers together (or perhaps just one or the other) can be used to distinguish, on the basis of time of emergence, between different sensitivities of carbonate export to OA.

[Figure]

So my recommendation is to conduct some more model simulations to assess sensitivity to these other changes, and if possible to also consider a study that combines both 230Th and alkalinity.

Other comments:

- The model does not include hydrothermal particles, which are important scavengers for Th (e.g. recent papers by Hayes) - Although the authors cite Heinze et al. 2006 as evidence for a lack of sensitivity to the choice of particle phases scavenging Th, it would be good to actually show that result in the context of the present study, as part of the sensitivity analysis. - I don't think figure 6 is necessary - Section plots would be more useful for the model-data comparison than the station by station comparisons

---

## Author Comment (AC1) · 13 Dec 2017

**Response to Anonymous Referee #1**

*(Referee's comments in italics.)*

Our response 1.0:

We would like to thank the referee for the very valuable comments and suggestions.

*This modelling study seeks to test whether the oceanic distribution of dissolved $^{230}$Th could serve as an indicator of reduced biogenic CaCO$_3$ formation as the ocean acidifies due to anthropogenic CO$_2$ emissions. It proposes that $^{230}$Th concentrations, particularly in the deep ocean, may be a more sensitive indicator of such change than direct observations of changing alkalinity in the surface ocean. In some ways, the modelling work described is a rather incremental advance relative to earlier work assessing the $^{230}$Th response to change in the CaCO$_3$:POC ratio presented in Heinze et al. 2006. The present manuscript, however, focuses specifically on testing how this ratio might be influenced by future ocean acidification, and whether this could be detected by $^{230}$Th measurements.*

*The ability to detect systematic change in the production of biogenic CaCO$_3$ in response to ocean acidification would be a useful tool, making this modelling endeavour potentially useful. The idea that $^{230}$Th may allow such detection is not intuitive, but interesting and worthy of consideration. So the general direction of this contribution is welcome. I am, however, unsure from the present manuscript that the utility of $^{230}$Th to assess CaCO$_3$ flux has been demonstrated.*

Our response 1.1:

This study here goes beyond the work of Heinze et al. (2006), where only the effect of an instantaneous switch in CaCO$_3$ rain ratio for one single grid point time series has been shown and only a few lines of text were devoted to ocean acidification (Figure 8 and pages 10-11 of Heinze et al. (2006)). In our study here, we investigate the global ocean $^{230}$Th reaction for a series of realistic CO$_2$ emission scenarios, we employ an improved model with respect to simulation of the CaCO$_3$:POC rain ratio pattern, have recalibrated the scavenging rate constants, and add an analysis of the time of emergence of a clearly identifiable signal in $^{230}$Th. Further, we test the $^{230}$Th reaction for different sensitivities of CaCO$_3$ to ocean acidification (based on the option as used in the study by (Ilyina et al., 2009). In addition, there seems some urgency to establish methods for detecting large-scale ocean acidification impacts as the respective integrated effect on ecosystems is not well known (see Gattuso et al. (2015), citation: "*Most studies have investigated the effects of ocean acidification on isolated organisms; far less is known about the effects on communities and ecosystems.*"). Therefore, our study here is fully justified. We address possible improvements of the assessment of CaCO$_3$ fluxes through $^{230}$Th below.

*1. Threshold for detection: The authors assume that detection of change in $^{230}$Th depends only on the analytical uncertainty of measurement. Measurements of $^{230}$Th in seawater shows significant spatial and temporal variation, however, which far exceed measurement uncertainty. Some of this variation reflects known processes, such as productivity or large-scale circulation, which cause consistent spatial patterns. But other variation is akin to 'noise', caused by seasonality of particle flux, eddying circulation, variability in boundary scavenging etc. To assess the possibility to detect change in the profile of $^{230}$Th due to variation in the composition of settling particles requires consideration of the natural variability of the $^{230}$Th field. One way to consider this might be to statistically compare closely spaced samples in the ever-growing observational $^{230}$Th dataset to assess small-scale natural variability. My guess is that a more realistic detection threshold is likely to be 2 to 3 times higher than the value assumed in this study. That would not prevent detection in the deep ocean (e.g. in Fig 9) but*

*would delay the date of detectability in that setting, and would prevent detection at shallower depths. Intuitively that seems realistic given that intermediate depths typically show quite large (and presently poorly explained) temporal changes in $^{230}$Th concentrations.*

Our response 1.2:

We agree with the referee that the detection levels shown in Figures 8, 9, and 10 would be the earliest possible (assuming that the preindustrial levels would be known as well). This has also been written in the text (page 10, line 22). If other factors than changes in $CaCO_3$ would occur this could change. However, Figures 8-10 show large-scale averages for entire oceans within the model world. It should be legitimate to show this earliest possible detection threshold for the average of a large region. Figures 8-10 do not make a judgement on how good an observing system should be to fully exploit the potential of $^{230}$Th to diagnose large-scale changes in $CaCO_3$ flux. Figures 8-10, however, demonstrate the potential of $^{230}$Th to detect such changes. For large-scale averages, the noise should cancel out. We plan to add a few sensitivity experiments to clarify the issue raised by the referee. Because we use an annual average constant velocity field, adding "natural variability" is not possible in our model set up in a dynamical sense (see also our response 1.3). This would require work on the dynamical physical model delivering the velocity field including synoptic forcing, new spin-ups of the circulation model as well as the biogeochemical model, and possibly new parameter tuning. We will explore simpler methods in order to address the point. We are currently thinking about two options. We could use a Monte-Carlo-sub-sampling method taking into account different length-scales away from the respect central model grid point and see how robust the signal for detection would remain. Another method would be to randomly perturb the nutrient uptake velocity ($V_{max}$) and the half saturation constant ($K_s$) in the Michaelis-Menten formulation for biogenic organic particle production. We also will try to find estimates on the area of influence for Eulerian time series stations (and how it may change with depth).

*2. Sensitivity to other changes: To be a useful monitor for $CaCO_3$ flux change, future $^{230}$Th concentrations must be more responsive to that process than to other possible changes. There is very little consideration in the manuscript of other likely controls on the $^{230}$Th distribution. These might include future changes in circulation driven by changing wind fields or freshwater inputs; changing productivity of organic carbon due to circulation changes; changing remineralisation of organic carbon due to altered $O_2$ levels; changing fluxes of silicate dust due to changing winds and continental aridity; or changing ballasting related to ecosystem change. I do not have an instinct about whether any such changes are likely to generate substantial changes in the $^{230}$Th field, but this seems a fundamental issue for a modelling study such as this one to address. Can the authors do more to assess whether $CaCO_3$ fluxes are really the dominant control on $^{230}$Th change? Or only one of several global changes that will alter the field?*

Our response 1.3:

Our paper focuses on $^{230}$Th as a tool for detecting $CaCO_3$ production changes. We have discussed the limitation of our approach on page 16, lines 10-22, including the use of a constant velocity field. We will extend this discussion in order to spell out the various uncertainty sources more clearly to the reader. For this we also will explore the possibility of further useful sensitivity experiments (as the one on randomly varying $V_{max}$ and $K_s$, see our response 1.2). We will choose a limited number of parameters that we will perturb and assess the effect on the marine $^{230}$Th distribution as compared to the effect of varying $CaCO_3$ fluxes, such as variations in the dust flux (i.e., the admixture of inert clay material from continental sources), natural variability in the rain ratios $POC:CaCO_3:BSi$ or changes in the particle specific scavenging. Potentially, we also could think of changing the velocity

field though this would be done through a kinematic and not dynamically consistent approach. Thus, the approach may be of limited explanatory power. In such an approach, we would combine the presently used field with a pseudo-glacial field of reduced overturning and by scaling such a combination with the meridional overturning variability time scale as found in simulations with a fully-fledged earth system model.

*3. Accuracy of the model: The $^{230}$Th model used is well established and has been thoroughly documented in the literature before, but there are some presentational issues in the present manuscript that limit the reader's ability to assess its prediction of future $^{230}$Th change:*

*i. Since Heinze 2006, there are significantly more $^{230}$Th observations, including long ocean sections (see* [*http://www.egeotraces.org/?group=Dissolved%20Natural%20Radionuclides,variable=Th%20230%20dissolved*](http://www.egeotraces.org/?group=Dissolved%20Natural%20Radionuclides,variable=Th%20230%20dissolved)*). It is now possible to directly compare modelled sections (e.g. Fig 3) with observations, and this should be done in this manuscript. Doing so reveals some quite important discrepancies, particularly in the deep ocean, which is being touted here as a sensitive indicator for changing CaCO$_3$ fluxes. These discrepancies include deviations related to scavenging at the seafloor and in MOR plumes. If these processes are not considered, the deep-ocean sensitivity of $^{230}$Th to downward particle flux may well be overestimated.*

Our response 1.4:

We intend to add a meridional Atlantic cross section of dissolved $^{230}$Th for both model and observations and discuss the discrepancies and their potential implications for diagnosing CaCO$_3$ flux changes through $^{230}$Th.

*ii. Perhaps I have misunderstood, but Figures 8-10 indicate that even the control run shows a significant change in deep $^{230}$Th, despite the lack of CaCO$_3$ change in this run. This is puzzling, and seems to suggest a problem with the long-term handling of $^{230}$Th in the model?*

Our response 1.5:

We explain this already in the submitted manuscript on page 11, lines 1-4: "For constant CaCO$_3$ production, the intermediate and deep water $^{230}$Th activities start to rise around year 2100 as well (see black curves in Figures 8-10). This effect is due to the increasing dissolution of CaCO$_3$ particles in the water column in parallel with downward mixing of waters that carry anthropogenic loads of dissolved organic carbon and hence subsurface and deep acidification." We will expand this paragraph in order to explain this more clearly. The effect becomes important only in areas, where anthropogenic carbon is mixed down quickly and induces a significant shoaling of the CaCO$_3$ saturation level and CaCO$_3$ lysocline as wall the Carbonate Compensation Depth through deep-water acidification. Most of the deep Pacific is not really influenced much by this within the 21$^{st}$ century.

*iii. Less significant, but it would also be good to see how the model predicts change as a profile or section, rather than as a time series at a single depth. As you go to greater depth in the ocean, the residence time of $^{230}$Th increases, so change might be slower, but the flux of organic carbon decreases so the influence of a CaCO$_3$ change will be more important. Seeing how such depth-related effects compete in the model would be interesting, and help to assess how realistic it is in representing the natural cycle.*

Our response 1.6:

We plan to add either the same section as mentioned in response 1.4 or representative profiles for subsequent time slices based on data from the additional sensitivity experiments. We will then discuss these diagrams for depth-related effects.

*Overall, these three concerns leave me unconvinced that this study is ready for publication. The idea of using $^{230}$Th to assess CaCO$_3$ fluxes is interesting, however, so I'd encourage the authors to seek to address these issues. A revised version of the work could then be a useful contribution.*

Our response 1.7:

We will address this issue in the revised version, please, see our responses 1.3-1.6.

*Other comments:*

*P4-6: The description of the model set up could be reduced, given that this is a previously described model, and that some aspects (e.g. C isotopes) are not relevant to this study.*

Our response 1.8:

In reviews of previous publications, where we omitted a detailed model description, the respective referees asked us to include a more detailed description so that readers would not have to read another (or more) articles in parallel. We, therefore, would like to keep the model description but will remove those elements, which are not relevant for this manuscript.

*P7: It's good to see the GEOTRACES data used, but the source of this data is strangely attributed. Neither de Baar nor Boyle were involved in collection of $^{230}$Th data. Please cite the relevant papers directly for this data (e.g. Hayes et al., Deng et al) in addition to the Mawji et al. paper.*

Our response 1.9:

We will add respective references to the paper – many thanks for pointing this out.

*Are any spatial changes expected because of changing rain-ratio? High latitude waters will decrease saturation faster than mid-latitude, so changes may be more acute there. I wonder if looking at the relative change in $^{230}$Th between regions may be a more sensitive indicator of the specific response to changing carbonate saturation than the general deep-ocean response?*

Our response 1.10:

This is an interesting metric for analysis – many thanks. We will look for such changes between selected regions and discuss this in the revised manuscript. See also our response 2.3 to Referee #2 concerning the comparison between different regions.

REFERENCES:

Gattuso, J. P., Magnan, A., Bille, R., Cheung, W. W. L., Howes, E. L., Joos, F., Allemand, D., Bopp, L., Cooley, S. R., Eakin, C. M., Hoegh-Guldberg, O., Kelly, R. P., Portner, H. O., Rogers, A. D., Baxter, J. M., Laffoley, D., Osborn, D., Rankovic, A., Rochette, J., Sumaila, U. R., Treyer, S., and Turley, C.: Contrasting futures for ocean and society from different anthropogenic CO2 emissions scenarios, Science, 349, ARTN aac4722, 10.1126/science.aac4722, 2015.

Heinze, C., Gehlen, M., and Land, C.: On the potential of Th-230, Pa-231, and Be-10 for marine rain ratio determinations: A modeling study, Global Biogeochemical Cycles, 20, Artn Gb2018, 10.1029/2005gb002595, 2006.

Ilyina, T., Zeebe, R. E., Maier-Reimer, E., and Heinze, C.: Early detection of ocean acidification effects on marine calcification, Global Biogeochemical Cycles, 23, Artn Gb1008, 10.1029/2008gb003278, 2009.

---

## Author Comment (AC2) · 13 Dec 2017

**Response to Anonymous Referee #2**

*(Referee's comments in italics.)*

Our response 2.0:

We would like to thank the referee for the very useful suggestions and comments.

*In this manuscript Heinze et al. explore the feasibility of using measurements of dissolved $^{230}$Th in the deep ocean to detect changes in calcification as a result of ocean acidification. The idea being it would be nice to be able to "check" the response of carbonate export/calcification to ocean acidification with observations of changes in both parameters. They argue that this approach is complementary to using measurements of alkalinity in the surface ocean to detect changes in calcification. This is an intriguing idea, especially the idea that surface changes are 'magnified' in the deep water through $^{230}$Th. And I fully agree with the authors' concluding sentence that the full potential of $^{230}$Th has yet to be realised. However, I have some reservations about this manuscript.*

*First, the idea is not actually new, having been proposed over 10 years ago by the first author (Heinze et al, 2006).*

Our response 2.1:

The idea was brought up by Heinze et al. (2006). This study follows up on it and pursues it in greater detail focusing now on the possibility of detecting changes in carbonate production through the monitoring of deep-water dissolved $^{230}$Th levels. See also our detailed response 1.1 in reply to Referee #1.

*More importantly, and particularly as the current work can be considered a 'follow up' on the initial work, this work is missing a key sensitivity analysis. Stepping back, the authors have shown that it's quite possible that changes in carbonate production and export will result in measureable changes in $^{230}$Th concentrations in the deep sea in coming decades or at least centuries. However, they have not adequately demonstrated that it will be possible to distinguish between the different 'parameterisations', or 'sensitivities' of calcification to OA, using the $^{230}$Th measurements, which is the premise of this work. In their experiments, where only $CO_2$ and calcification change, yes there are clear differences in the time of emergence for the different parameterisations. However, in the real world there will almost certainly be changes in POC, $CaCO_3$ and opal that are independent of OA, driven by changes in stratification, temperature, dust deposition, even anthropogenic nutrient input. Do the different Th trajectories and times of emergence stay distinctly different when you add in these other changes? Furthermore, ocean circulation will likely change and that will change $^{230}$Th distributions independent of changes in particle flux and composition. Again, can you distinguish between different OA sensitivities once you factor in a range of ocean circulations? It all comes down to the sensitivity of $^{230}$Th distributions to OA compared to their sensitivity to other perturbations.*

Our response 2.2:

We intend to remedy this point by adding additional sensitivity experiments as also suggested by Referee #1. We think that our study has a value even without such additional sensitivity experiments, but agree that these would strengthen the manuscript. Please, see our responses 1.2 and 1.3 in reply to Referee #1.

*The authors mention that changes in deep ocean $^{230}$Th concentration emerge at about the same time as changes in alkalinity, and point out that the two tracers offer complementary information. I think this is an excellent point. My feeling is a study that combines these two tracers, together with a*

*sensitivity analysis as suggested here, could potentially be quite useful in showing how the two tracers together (or perhaps just one or the other) can be used to distinguish, on the basis of time of emergence, between different sensitivities of carbonate export to OA.*

Our response 2.3:

This is an interesting point – thank you. We will address this issue by evaluating the model results through consideration of simultaneous changes in alkalinity and $^{230}$Th at various regions in connection with the regional assessment as suggested by Referee #1 (see also our reply 1.10 to Referee #1).

*So my recommendation is to conduct some more model simulations to assess sensitivity to these other changes, and if possible to also consider a study that combines both $^{230}$Th and alkalinity.*

Our response 2.4:

See our responses 2.2 and 2.3 for this.

*Other comments:*

*- The model does not include hydrothermal particles, which are important scavengers for Th (e.g. recent papers by Hayes)*

Our response 2.5:

We will discuss this issue with additional graphical material. Please, see our response 1.4 to Referee #1.

*- Although the authors cite Heinze et al. 2006 as evidence for a lack of sensitivity to the choice of particle phases scavenging Th, it would be good to actually show that result in the context of the present study, as part of the sensitivity analysis.*

Our response 2.6:

We will repeat a few model runs with a change in particle specific scavenging and discuss whether this issue is of relevance for the results presented here. (See also our response 1.3 to Referee #1).

*- I don't think figure 6 is necessary - Section plots would be more useful for the model-data comparison than the station by station comparisons.*

Our response 2.7:

We would like to keep Figure 6, as not every biogeochemical trace element researcher is fully familiar with the greenhouse gas scenarios. Plots of observations and model results will be added for an Atlantic meridional cross section (see also our response to Referee #1).

REFERENCE:

Heinze, C., Gehlen, M., and Land, C.: On the potential of Th-230, Pa-231, and Be-10 for marine rain ratio determinations: A modeling study, Global Biogeochemical Cycles, 20, Artn Gb2018, 10.1029/2005gb002595, 2006.

---

## Author Response (AR2)

**RESPONSE TO THE REFEREES AND REPORT ON CHANGES MADE IN MANUSCRIPT bg-2017-305**

**Additional/general changes made (*for direct responses to Referees #1 and #2, see further below*):**

MODEL RUNS:

All model runs have been repeated on the basis of a slightly improved preindustrial reference run. The newly introduced comparison with the observed Atlantic dissolved $^{230}$Th section revealed somewhat too low values in the upper 1000 m. The changes with respect to the original reference run are: A stronger reduction of biological particle production through reducing $V_{max}$ (nutrient uptake velocity) increasing see ice thickness, a reduction of the clay flux for $^{230}$Th scavenging (from atmospheric dust input which may have been overestimated for the Atlantic originally) to 25% (a sensitivity experiment with 100% dust flux was also added see below), and a change in the equilibrium coefficient $k_d$ between dissolved and particle attached phases taking particle specific scavenging rates into account (following Hayes et al., Marine Chemistry 170 (2015) 49–60).

As compared to the original manuscript, a suite of sensitivity experiments has been added to the analysis. These experiments address:

A) With respect to the new preindustrial control run:

- A general slight increase in the scavenging equilibrium coefficient $k_d$.

- A strong increase of the scavenging equilibrium coefficient $k_d$ in the grid cells directly above the ocean floor (to test the effect of resuspension of particles and hydrothermal vents).

- An increase of the clay flux to 100% of the prescribed dust input at the ocean surface.

- Scavenging to $CaCO_3$ and clay only, not to POC.

- Scavenging to clay and POC only, not to $CaCO_3$.

B) With respect to the RCP8.5 moderate calcification projection (changes scaled by the decadal change resulting from the fully-fledged Earth system model NorESM):

- A reduction/variation in ocean overturning.

- A reduction/variation in $V_{max}$ (nutrient uptake velocity).

- A reduction/variation in particle sinking velocity.

- A simultaneous reduction/variation of overturning, $V_{max}$, and particle sinking velocity.

C) With respect to the RCP8.5 extreme calcification projection (changes scaled by the decadal change resulting from the fully-fledged Earth system model NorESM):

- A simultaneous reduction/variation of overturning, $V_{max}$, and particle sinking velocity.

FIGURES:

In order to account for the wishes of the referees and to keep a good balance between text and figures, we introduced some new figures and modified some original figures. A part of the figures was moved to a newly established Supplementary material.

Figure 3: Newly drawn with new reference run and using a somewhat different "cruise track" for the Atlantic section.

Figure 4: Newly added Atlantic section from observations and model minus observations section.

Original Figure 4 has become new Figure 5: The single station profiles have been removed (because we added the observed section), and the scatter diagram has been redrawn.

Original Figure 5 has become new Figure 6: The biological production maps have been redrawn using the new reference run data.

New Figure S1: Atlantic and Pacific meridional cross sections have been added for the scavenging equilibrium coefficient $k_d$ in the Supplementary Material.

Newly added Figure 7: Cross sections documenting the results of the sensitivity experiments with respect to the preindustrial control run.

Original Figure 6 on RCP scenarios has been moved to Figure S2 in the Supplementary Material.

Original Figure 7 has become new Figure 8 and was redrawn using the data from the new runs.

Original Figure 8 has become new Figure S3 in the Supplementary Material: Figure S3 shows a higher sensitivity in dissolved $^{230}$Th with at 700 m depth with respect to a decrease in $CaCO_3$ production due to the revised scavenging formulation. The y-axis resolution was changed so that the cross over points for the theoretical detection level are better resolved. The control run values are now shown for the respective control run using the RCP atmospheric $CO_2$ concentrations, but no change in $CaCO_3$ production (originally only the RCP8.5 control run values have been shown; this has been corrected now).

Original Figure 9 has been redrawn: The y-axis resolution was changed so that the cross over points for the theoretical detection level are better resolved. The control run values are now shown for the respective control run using the RCP atmospheric $CO_2$ concentrations, but no change in $CaCO_3$ production (originally only the RCP8.5 control run values have been shown; this has been corrected now).

Original Figure 10 has become new Figure S4 in the Supplementary Material: The y-axis resolution was changed so that the cross over points for the theoretical detection level are better resolved. The control run values are now shown for the respective control run using the RCP atmospheric $CO_2$ concentrations, but no change in $CaCO_3$ production (originally only the RCP8.5 control run values have been shown; this has been corrected now).

New Figure 10 replaces the original Figures 11 and 12: Figure 10 was draw using the results from the revised model runs.

Newly added Figure 11: Cross sections on dissolved $^{230}$Th have been added which show the change due to decadal variability in various model parameters relative to a scenario run with calcification reduction effect only. The data are from the new sensitivity experiments concerning the projections.

Newly added Figure 12: The time of signal emergence has been shown now also for the runs with simultaneous decadal changes in model parameters.

Newly added Figure 13: Maps of surface and deep changes in alkalinity and dissolved $^{230}$Th during 2040-2010 as for the RCP8.5 projection have been drawn using moderate calcification changes. This illustrates the difference in reactions to $CaCO_3$ production changes for the two tracers alkalinity and dissolved $^{230}$Th.

**Response to Anonymous Referee #1**

*(Referee's comments in italics. Our references to figures, page/line numbers etc. in the response to the referees refer to the originally submitted manuscript except where it is explicitly stated otherwise.)*

Our response 1.0:

We would like to thank the referee for the very valuable comments and suggestions.

*This modelling study seeks to test whether the oceanic distribution of dissolved [230]Th could serve as an indicator of reduced biogenic $CaCO_3$ formation as the ocean acidifies due to anthropogenic $CO_2$ emissions. It proposes that [230]Th concentrations, particularly in the deep ocean, may be a more sensitive indicator of such change than direct observations of changing alkalinity in the surface ocean. In some ways, the modelling work described is a rather incremental advance relative to earlier work assessing the [230]Th response to change in the $CaCO_3$:POC ratio presented in Heinze et al. 2006. The present manuscript, however, focuses specifically on testing how this ratio might be influenced by future ocean acidification, and whether this could be detected by [230]Th measurements.*

*The ability to detect systematic change in the production of biogenic $CaCO_3$ in response to ocean acidification would be a useful tool, making this modelling endeavour potentially useful. The idea that [230]Th may allow such detection is not intuitive, but interesting and worthy of consideration. Therefore, the general direction of this contribution is welcome. I am, however, unsure from the present manuscript that the utility of [230]Th to assess $CaCO_3$ flux has been demonstrated.*

Our response 1.1:

This study here goes beyond the work of Heinze et al. (2006), where only the effect of an instantaneous switch in $CaCO_3$ rain ratio for one single grid point time series has been shown and only a few lines of text were devoted to ocean acidification (Figure 8 and pages 10-11 of Heinze et al. (2006)). In our study here, we investigate the global ocean [230]Th reaction for a series of realistic $CO_2$ emission scenarios, we employ an improved model with respect to simulation of the $CaCO_3$:POC rain ratio pattern, have recalibrated the scavenging rate constants, and add an analysis of the time of emergence of a clearly identifiable signal in [230]Th. Further, we test the [230]Th reaction for different sensitivities of $CaCO_3$ to ocean acidification (based on the option as used in the study by (Ilyina et al., 2009). In addition, there seems some urgency to establish methods for detecting large-scale ocean acidification impacts as the respective integrated effect on ecosystems is not well known (see Gattuso et al. (2015), citation: "*Most studies have investigated the effects of ocean acidification on isolated organisms; far less is known about the effects on communities and ecosystems.*"). Therefore, our study here is fully justified. We address possible improvements of the assessment of $CaCO_3$ fluxes through [230]Th below.

*1. Threshold for detection: The authors assume that detection of change in [230]Th depends only on the analytical uncertainty of measurement. Measurements of [230]Th in seawater shows significant spatial and temporal variation, however, which far exceed measurement uncertainty. Some of this variation reflects known processes, such as productivity or large-scale circulation, which cause consistent spatial patterns. But other variation is akin to 'noise', caused by seasonality of particle flux, eddying circulation, variability in boundary scavenging etc. To assess the possibility to detect change in the profile of [230]Th due to variation in the composition of settling particles requires consideration of the natural variability of the [230]Th field. One way to consider this might be to statistically compare closely spaced samples in the ever-growing observational [230]Th dataset to assess small-scale natural*

*variability. My guess is that a more realistic detection threshold is likely to be 2 to 3 times higher than the value assumed in this study. That would not prevent detection in the deep ocean (e.g. in Fig 9) but would delay the date of detectability in that setting, and would prevent detection at shallower depths. Intuitively that seems realistic given that intermediate depths typically show quite large (and presently poorly explained) temporal changes in $^{230}$Th concentrations.*

Our response 1.2:

We agree with the referee that the detection levels shown in Figures 8, 9, and 10 would be the earliest possible (assuming that the preindustrial levels would be known as well). This has also been written in the text (page 10, line 22). If other factors than changes in $CaCO_3$ would occur this could change. However, Figures 8-10 show large-scale averages for entire oceans within the model world. It should be legitimate to show this earliest possible detection threshold for the average of a large region. Figures 8-10 do not make a judgement on how good an observing system should be to fully exploit the potential of $^{230}$Th to diagnose large-scale changes in $CaCO_3$ flux. Figures 8-10, however, demonstrate the potential of $^{230}$Th to detect such changes. For large-scale averages, the noise should cancel out.  We have carried out a number of additional sensitivity experiments concerning the reference run without anthropogenic rising $CO_2$ and with rising $pCO_2$. These sensitivity experiments are described under section 4 (renamed to "4 Control run, scenario experiments, and sensitivity experiments") and discussed in section 5. Table 2 has been added as an overview about the various experiments. The impact of the sensitivity experiments is illustrated in the new figures 7, 11, and 13 and discussed in sections 5 and 6.

*2. Sensitivity to other changes: To be a useful monitor for CaCO$_3$ flux change, future $^{230}$Th concentrations must be more responsive to that process than to other possible changes. There is very little consideration in the manuscript of other likely controls on the $^{230}$Th distribution. These might include future changes in circulation driven by changing wind fields or freshwater inputs; changing productivity of organic carbon due to circulation changes; changing remineralisation of organic carbon due to altered O$_2$ levels; changing fluxes of silicate dust due to changing winds and continental aridity; or changing ballasting related to ecosystem change. I do not have an instinct about whether any such changes are likely to generate substantial changes in the $^{230}$Th field, but this seems a fundamental issue for a modelling study such as this one to address. Can the authors do more to assess whether CaCO$_3$ fluxes are really the dominant control on $^{230}$Th change? Or only one of several global changes that will alter the field?*

Our response 1.3:

Our paper focuses on $^{230}$Th as a tool for detecting $CaCO_3$ production changes. We have discussed the limitation of our approach on page 16, lines 10-22, including the use of a constant velocity field. In the revised manuscript, we have added a series of sensitivity experiments also with respect to the preindustrial reference run (runs P1-P5, see new Table 2). These runs address uncertainties in the equilibrium coefficient $k_d$ in general and specifically in view of hydrothermal vents, the strength of the clay flux, and the effect of omitting POC or $CacO_3$ as carrier phases for $^{230}$Th. The experiments are described in sections 4 and discussed in section 6. The new figure 7 was added illustrating the results. In addition, the sensitivity experiments relative to the projections (S1-S5, Table 2) show the effects of circulation changes, changes in nutrient uptake velocity, and particle sinking speed. The set-up for these experiments S1-S5 mimicking interannual/decadal variability through scaling of parameter changes to the overturning change time series from a fully-fledged Earth system model is presented in section 4.

*3. Accuracy of the model: The $^{230}$Th model used is well established and has been thoroughly documented in the literature before, but there are some presentational issues in the present manuscript that limit the reader's ability to assess its prediction of future $^{230}$Th change:*

*i. Since Heinze 2006, there are significantly more $^{230}$Th observations, including long ocean sections (see http://www.egeotraces.org/?group=Dissolved%20Natural%20Radionuclides,variable=Th%20230%20dissolved). It is now possible to directly compare modelled sections (e.g. Fig 3) with observations, and this should be done in this manuscript. Doing so reveals some quite important discrepancies, particularly in the deep ocean, which is being touted here as a sensitive indicator for changing CaCO$_3$ fluxes. These discrepancies include deviations related to scavenging at the seafloor and in MOR plumes. If these processes are not considered, the deep-ocean sensitivity of $^{230}$Th to downward particle flux may well be overestimated.*

Our response 1.4:

We have added a meridional Atlantic cross section of dissolved $^{230}$Th for both model and observations and discussed the discrepancies and their potential implications for diagnosing CaCO$_3$ flux changes through $^{230}$Th. To this end, new Figures 3, 4, and 7 have been added and discussed. Because the reference run of the original manuscript showed somewhat too low $^{230}$Th values in the upper North Atlantic (representing a large relative error) we reformulated and retuned the model, and repeated all model experiments. The changes in the new model are a partial further reduction of particle production under ice (scaled with the ice thickness), using only 25% of the clay flux for scavenging, and adjusting the k$_d$ value with respect to particle composition. The model description has been updated accordingly.

*ii. Perhaps I have misunderstood, but Figures 8-10 indicate that even the control run shows a significant change in deep $^{230}$Th, despite the lack of CaCO$_3$ change in this run. This is puzzling, and seems to suggest a problem with the long-term handling of $^{230}$Th in the model?*

Our response 1.5:

We explain this already in the submitted manuscript on page 11, lines 1-4: "For constant CaCO$_3$ production, the intermediate and deep water $^{230}$Th activities start to rise around year 2100 as well (see black curves in Figures 8-10). This effect is due to the increasing dissolution of CaCO$_3$ particles in the water column in parallel with downward mixing of waters that carry anthropogenic loads of dissolved organic carbon and hence subsurface and deep acidification." We expanded the text in order to explain this more clearly: "For constant CaCO$_3$ production, the intermediate and deep water $^{230}$Th activities start to rise around year 2100 as well (see black curves in Figures S3, 9, and S4). This effect is due to the increasing dissolution of CaCO$_3$ particles in the water column in parallel with downward mixing of waters that carry anthropogenic loads of dissolved organic carbon and hence subsurface and deep acidification. The effect is most important in areas, where anthropogenic carbon is mixed down quickly and induces a significant shoaling of the CaCO$_3$ saturation level and CaCO$_3$ lysocline as well the Carbonate Compensation Depth through deep-water acidification. Parts of the deep Pacific are not as much influenced by this as compared to the Atlantic within the 21st century. The control run in the figures does not represent the reference run with constant preindustrial atmospheric pCO$_2$ but the run with constant CaCO$_3$ production and rising pCO$_2$ according to the RCP scenarios."

*iii. Less significant, but it would also be good to see how the model predicts change as a profile or section, rather than as a time series at a single depth. As you go to greater depth in the ocean, the*

*residence time of $^{230}$Th increases, so change might be slower, but the flux of organic carbon decreases so the influence of a CaCO$_3$ change will be more important. Seeing how such depth-related effects compete in the model would be interesting, and help to assess how realistic it is in representing the natural cycle.*

Our response 1.6:

We have added respective cross sections on how the Atlantic and Pacific meridional $^{230}$Th distribution changes if we assume decadal variability in selected parameters relative to a case with constant biological production (new Figure 11). These cross sections show that also the effects of decadal variability will increase with depth for changes in biological carbon cycling and strength of ocean overturning.

*Overall, these three concerns leave me unconvinced that this study is ready for publication. The idea of using $^{230}$Th to assess CaCO$_3$ fluxes is interesting, however, so I'd encourage the authors to seek to address these issues. A revised version of the work could then be a useful contribution.*

Our response 1.7:

Our results for this revised version show that for strong changes in CaCO$_3$ production, the $^{230}$Th signal in deep waters still is a promising respective indicator. For more moderate changes in CaCO$_3$ production, the $^{230}$Th signal in deep waters may be masked by other effects such as changes in particle sinking velocity and circulation. Sections 5 and 6, and the abstract have been updated respectively.

*Other comments:*

*P4-6: The description of the model set up could be reduced, given that this is a previously described model, and that some aspects (e.g. C isotopes) are not relevant to this study.*

Our response 1.8:

In reviews of previous publications, where we omitted a detailed model description, the respective referees asked us to include a more detailed description so that readers would not have to read another (or more) articles in parallel. We, therefore, would like to keep the model description but will remove those elements, which are not relevant for this manuscript. We have added a description of the revised scavenging formulation for the experiments shown in this revised version.

*P7: It's good to see the GEOTRACES data used, but the source of this data is strangely attributed. Neither de Baar nor Boyle were involved in collection of $^{230}$Th data. Please cite the relevant papers directly for this data (e.g. Hayes et al., Deng et al) in addition to the Mawji et al. paper.*

Our response 1.9:

We have corrected the references for the GEOTRACES data set – many thanks for pointing this out.

*Are any spatial changes expected because of changing rain-ratio? High latitude waters will decrease saturation faster than mid-latitude, so changes may be more acute there. I wonder if looking at the relative change in $^{230}$Th between regions may be a more sensitive indicator of the specific response to changing carbonate saturation than the general deep-ocean response?*

Our response 1.10:

This is an interesting metric for analysis – many thanks. We have looked at this. Indeed we could detect a faster rain ratio change $CaCO_3$/POC at high latitudes than in lower latitudes. However, this signal was not directly translated into a simple deep $^{230}$Th change pattern.

**Response to Anonymous Referee #2**

*(Referee's comments in italics. Our references to figures, page/line numbers etc. in the response to the referees refer to the originally submitted manuscript except where it is explicitly stated otherwise.)*

Our response 2.0:

We would like to thank the referee for the very useful suggestions and comments.

*In this manuscript Heinze et al. explore the feasibility of using measurements of dissolved $^{230}$Th in the deep ocean to detect changes in calcification as a result of ocean acidification. The idea being it would be nice to be able to "check" the response of carbonate export/calcification to ocean acidification with observations of changes in both parameters. They argue that this approach is complementary to using measurements of alkalinity in the surface ocean to detect changes in calcification. This is an intriguing idea, especially the idea that surface changes are 'magnified' in the deep water through $^{230}$Th. And I fully agree with the authors' concluding sentence that the full potential of $^{230}$Th has yet to be realised. However, I have some reservations about this manuscript.*

*First, the idea is not actually new, having been proposed over 10 years ago by the first author (Heinze et al, 2006).*

Our response 2.1:

The idea was brought up by Heinze et al. (2006). This study follows up on it and pursues it in greater detail focusing now on the possibility of detecting changes in carbonate production through the monitoring of deep-water dissolved $^{230}$Th levels. See also our detailed response 1.1 in reply to Referee #1.

*More importantly, and particularly as the current work can be considered a 'follow up' on the initial work, this work is missing a key sensitivity analysis. Stepping back, the authors have shown that it's quite possible that changes in carbonate production and export will result in measureable changes in $^{230}$Th concentrations in the deep sea in coming decades or at least centuries. However, they have not adequately demonstrated that it will be possible to distinguish between the different 'parameterisations', or 'sensitivities' of calcification to OA, using the $^{230}$Th measurements, which is the premise of this work. In their experiments, where only $CO_2$ and calcification change, yes there are clear differences in the time of emergence for the different parameterisations. However, in the real world there will almost certainly be changes in POC, $CaCO_3$ and opal that are independent of OA, driven by changes in stratification, temperature, dust deposition, even anthropogenic nutrient input. Do the different Th trajectories and times of emergence stay distinctly different when you add in these other changes? Furthermore, ocean circulation will likely change and that will change $^{230}$Th distributions independent of changes in particle flux and composition. Again, can you distinguish between different OA sensitivities once you factor in a range of ocean circulations? It all comes down to the sensitivity of $^{230}$Th distributions to OA compared to their sensitivity to other perturbations.*

Our response 2.2:

We have added two sets of additional sensitivity experiments (a) with respect to the preindustrial reference run (experiments P1-P5, new Table 2) and (b) relative to the projections including time dependent change (experiments S1-S4, new Table 2). We think that our study has a value even without such additional sensitivity experiments, but agree that these experiments strengthen the manuscript. Please, see our detailed responses 1.2, 1.3, and 1.4 in reply to Referee #1.

*The authors mention that changes in deep ocean $^{230}$Th concentration emerge at about the same time as changes in alkalinity, and point out that the two tracers offer complementary information. I think this is an excellent point. My feeling is a study that combines these two tracers, together with a sensitivity analysis as suggested here, could potentially be quite useful in showing how the two tracers together (or perhaps just one or the other) can be used to distinguish, on the basis of time of emergence, between different sensitivities of carbonate export to OA.*

Our response 2.3:

This is an interesting point – thank you. We have tried to identify a systematic interdependence between alkalinity changes and $^{230}$Th changes, but so far could not find out how this could possibly lead to a new tracer for acidification impacts. We have added the new Figure 13 to illustrate this and describe the idea/issue in the discussion.

*So my recommendation is to conduct some more model simulations to assess sensitivity to these other changes, and if possible to also consider a study that combines both $^{230}$Th and alkalinity.*

Our response 2.4:

See our responses 2.2 and 2.3 for this.

*Other comments:*

*- The model does not include hydrothermal particles, which are important scavengers for Th (e.g. recent papers by Hayes)*

Our response 2.5:

Regionally too high modelled $^{230}$Th deep water values in the Atlantic section have been found (new Figure 4). We have added a sensitivity experiment, where we increased the equilibrium coefficient for $k_d$ for partitioning between dissolved and particle attached phases in the grid cells directly above the ocean floor. According to this, adding hydrothermal vents could be a means of reducing the discrepancy between model results and observations. See also the new Figure 7 and the updated discussion section.

*- Although the authors cite Heinze et al. 2006 as evidence for a lack of sensitivity to the choice of particle phases scavenging Th, it would be good to actually show that result in the context of the present study, as part of the sensitivity analysis.*

Our response 2.6:

We have added sensitivity experiments with omitting the scavenging to (a) POC, and (b) CaCO3, and further have carried out a run with changes in the clay flux for scavenging. Sections 4 and 5 have been updated accordingly and the new Figure 7 has been inserted.

*- I don't think figure 6 is necessary - Section plots would be more useful for the model-data comparison than the station by station comparisons.*

Our response 2.7:

We have kept Figure 6, as not every biogeochemical trace element researcher is fully familiar with the greenhouse gas scenarios, but moved Figure 6 to the Supplementary Material as Figure S2. Plots of observations and model results have been added for an Atlantic meridional cross section (see also our response to Referee #1).

[revised manuscript text omitted]

**Supplementary Material:**

[Figure]

**Figure S1:** Meridional cross sections of the scavenging equilibrium coefficient $k_d$ as used for the control runs and the standard future scenarios. (a) Atlantic. (b) Pacific.

[Figure]

**Figure S2:** $CO_2$ concentrations according to the Representative Concentration Pathways (RCPs, van Vuuren et al., 2011) as prescribed in the predictive model runs.

[Figure]

**Figure S3:** Time series for the evolution of mean Atlantic (left column) and Pacific (right column) dissolved $^{230}$Th concentrations at 700 m under the different scenarios for reduction of calcification under high $CO_2$ (unit: dpm/1000l). From top to bottom for greenhouse gas scenarios RCP8.5. RCP 6.0, RCP4.5, and RCP2.6. The orange line indicates the theoretical detection limit for changes with respect with respect to the pre-industrial.

[Figure]

**Figure S4:** Time series for the evolution of mean Atlantic (left column) and Pacific (right column) dissolved $^{230}$Th concentrations at 4000 m under the different scenarios for reduction of calcification under high $CO_2$ (unit: dpm/1000l). From top to bottom for greenhouse gas scenarios RCP8.5. RCP 6.0, RCP4.5, and RCP2.6. The orange line indicates the theoretical detection limit for changes with respect with respect to the pre-industrial.